



# Scale-dependency in modeling nivo-glacial hydrological systems: the case of the Arolla basin, Switzerland

Anne-Laure Argentin[1], Pascal Horton[2], Bettina Schaefli[2], Jamal Shokory[3], Felix Pitscheider[1], Leona Repnik[3], Mattia Gianini[3], Simone Bizzi[4], Stuart Lane[3], and Francesco Comiti[1,5]

[1]Faculty of Agricultural, Environmental and Food Sciences, Free University of Bozen-Bolzano, Italy
[2]Institute of Geography, and Oeschger Center on Climate Change Research, University of Bern, Bern Switzerland, Switzerland
[3]Institute of Earth Surface Dynamics, University of Lausanne, Switzerland
[4]Department of Geosciences, University of Padova, Italy
[5]Department of Land, Environment, Agriculture and Forestry, University of Padova, Italy

**Correspondence:** Anne-Laure Argentin (aargentin@unibz.it)

**Abstract.** Hydrological modeling in alpine catchments poses unique challenges due to the complex interplay of meteorological, topographical, glaciological and streamflow generation factors. A significant issue arises from the limited availability of streamflow data due to the scarcity of high-elevation gauging stations. Consequently, there is a pressing need to assess whether streamflow models that are calibrated with moderate-elevation datasets can be effectively transferred to higher-elevation catchments, notwithstanding differences in the relative importance of different streamflow-generation processes. Here, we investigate the spatial transferability of hydrological model parameters within a semi-lumped modeling framework. We focus on evaluating the model transferability from the main catchment to nested and neighboring subcatchments in the Arolla valley, southwestern Swiss Alps. We use the Hydrobricks modeling framework to simulate streamflow patterns, implementing three variants of a temperature-index snow- and ice melt model (the classical degree-day, aspect-related, and Hock's temperature index). Through a comprehensive analysis of streamflow simulations, benchmark metrics consisting of bootstrapped discharge series, and model performance, we demonstrate that robust parameter transferability and accurate streamflow simulation are possible across diverse spatial scales. This finding is conditional upon the used melt model, with melt models using more spatial information leading to convergence of the model parameters until there is an onset of overparameterization.

## 1 Introduction

Understanding the driving factors of nivo-glacial streamflow regimes is essential for managing high alpine catchments and their water resources under global change. With ongoing warming, the long-, intermediate- and short-term storage capacities of alpine glaciers are impacted (Jansson et al., 2003; Huss et al., 2008), and high alpine catchments may transition from nivo-glacial streamflow regimes to strictly nival regimes (Horton et al., 2006). Currently, alpine glaciated catchments and downstream areas receive a strong surplus of meltwater from snow during spring and early summer (Penna et al., 2017; Engel et al., 2019; Zuecco et al., 2019), gradually switching to glacier meltwater towards the end of summer. The timing and amount of snow- and glaciermelt water is strongly impacted by warming and related glacier retreat, making changes in





streamflow regimes critical (e.g., Singh and Kumar, 1997; Bradley et al., 2006). These changes in streamflow regimes and runoff generation characteristics can also have important consequences in terms of sediment transport, hydropower production (Gabbud et al., 2016), flood prediction and ecology (Tague et al., 2020).

However, high alpine catchments often lack discharge monitoring stations due to their sparse population and difficulty of access. In highly glacierized catchments (i.e. glacial cover > 50%), there are very few gauging stations that provide reliable and long-term streamflow records. This makes attributing historical changes in streamflow regimes to glacial sources challenging and inevitably requires recourse to modeling, not just to predict the future but also to understand the past.

     Hydrological models are commonly classified into distributed, semi-distributed, semi-lumped and lumped models (Horton
et al., 2022). Distributed models compute the storage and mobilization of water at the pixel scale, with parameters that vary in space (fully distributed) or are partially kept constant (semi-distributed). Semi-lumped models define areas of interest based on relevant physical parameters (e.g., elevation, aspect, stream network topology), while lumped models consider the catchment as a single unit with no spatial discretization. The advantage and popularity of (semi-)lumped models should not be reduced to their computational efficiency, which enables fast and multiple model runs. They represent an optimal level of model complex-
ity with respect to available input and output data from a downward model development perspective (Sivapalan et al., 2003); they furthermore operate at a scale at which averaging of small-scale processes enables a reliable representation of dominant hydrological processes (Clark et al., 2016).

     However, one of the main drawbacks of (semi-)lumped models is that streamflow can only be modeled reliably at the selected control points (outlets) for which the model parameters have been calibrated against observed streamflow. Simulated
streamflow at other locations within or near the catchment might not reliably represent the actual system dynamics. In other words, the calibrated parameters might not be transferrable to other locations of the stream network (subcatchments) within the system. This difficulty is exacerbated in catchments with strong process gradients and spatial heterogeneity, where the complex spatial averaging complicates the extraction of specific process responses at smaller scales, which is typically the case in glaciated catchments.

Parameter regionalization techniques (Guo et al., 2021) in hydrological modeling have been developed to facilitate the transfer of model parameters from gauged to ungauged locations (e.g., Mosley, 1981; Abdulla and Lettenmaier, 1997; Bardossy and Singh, 2008).

     Regionalization methods can be divided into two categories (Samaniego et al., 2010): post-regionalization and simultaneous regionalization. Postregionalization methods calibrate a model in several basins independently and then statistically link the
calibrated model parameters to basin predictors (e.g., mean catchment elevation, stream network density, geology, areal proportion of porous aquifers) using a transfer function (e.g., Abdulla and Lettenmaier, 1997; Seibert, 1999; Parajka et al., 2005; Wagener and Wheater, 2006). Simultaneous regionalizations aim to calibrate model parameters for several basins while taking into account transfer functions that link model parameters to catchment characteristics (e.g., Hundecha and Bárdossy, 2004; Götzinger and Bárdossy, 2007; Fernandez et al., 2000; Troy et al., 2008). The second category of methods was developed to
add additional spatial constraints to the calibration of the parameters and avoid artifacts of the optimization algorithm.





In all these methods, the need to define a function that links catchment characteristics and model parameters is subject to additional uncertainties. The number of parameter regionalization studies remains small (Horton et al., 2022), and the spatial transfer of model parameters is still a crucial topic for the prediction of streamflow in catchments without observed streamflow (Guo et al., 2021).

Spatial parameter transfer is particularly challenging in data-sparse high-elevation catchments where glacier melt, interannual snow storage and highly uncertain precipitation input and evapotranspiration output can lead to considerable parameter biases (Schaefli and Huss, 2011). A particular challenge in such catchments is the estimation of snow and glacier melt contributions, which, for practical, data reasons, is often limited to the use of temperature-index melt models (TI) that link melt rates to air temperature (Eq. 1, Rango and Martinec, 1995):

$$
M_{\mathrm{TI}}(t) = \begin{cases} a_j(T_a(t) - T_T) & : T_a(t) > T_T \quad \text{with } j \in \text{snow, ice} \\ 0 & : T_a(t) \leq T_T \end{cases} \tag{1}
$$

where $M_{\mathrm{TI}}(t)$ is the melt rate at time step $t$ (mm d$^{-1}$), $a_j$ the degree day factor for ice or snow (mm d$^{-1}$ °C$^{-1}$), $T_a$ is the air temperature and $T_T$ is the threshold melt temperature. Several declinations were developed from this classical temperature-index melt model.

Although it is commonly admitted that TI models present a good option for extrapolation to larger scales because of the consistency of temperature over large areas (Frenierre and Mark, 2014), the spatial transferability of the related parameters calibrated at the outlet of a catchment to the outlet of a neighboring catchment exhibiting different characteristics (elevation, aspect, glacial cover) can be questioned (Gabbi et al., 2014; Samaniego et al., 2010), but has been rarely investigated. This challenge was exemplified for nested catchments by Comola et al. (2015), who found significant variability in the calibrated degree-day factors for small catchments ($< 7$ km$^2$) due to the correlation of aspects when using a simple temperature-index model.

In this study, we investigate the transferability of the melt and runoff calibrated parameters between subcatchments and neighboring catchments depending on the melt model and the performance criterion. To do this, we calibrate our model on seven catchments, then take the parameters of the largest catchments to transfer them to its three nested watersheds and its three other neighboring catchments. We then analyze the loss of accuracy linked to the transfer of parameters. We carry out this operation with three temperature-index melt models of increasing complexity, and try answering the question: Could incorporating additional spatial information into the model increase its spatial transferability?

## 2 Study area: the upper Arolla river basin and its subcatchments

### 2.1 Presentation of the study area

We use data (Table 3) from the Arolla river basin located in the south-western Swiss Alps (Fig. 1). A local hydropower company was willing to provide 15-minute resolution streamflow recordings of very high quality given strict regulatory requirements for monitoring water use (Lane and Nienow, 2019). The Bertol Inférieur (BI) gauging station is fed by water draining from





**Table 1.** Catchments used in the simulations, and their properties

| Catchment | Abbre- viation | Area (km2) | Elevation (m) | | | Mean slope | Main as- pect | Glacier covered catchm. | Debris covered glacier | Type |
|---|---|---|---|---|---|---|---|---|---|---|
| | | | Mean | Min | Max | | | | | |
| Bertol Inférieur | BI | 26.0 | 3063 | 2183 | 3722 | 28.7 | NW | 38.5% | 9.9% | Main |
| Haut Glacier d'Arolla | HGDA | 13.2 | 3014 | 2582 | 3677 | 29.5 | NW | 32.0% | 16.3% | Nested |
| Tsijiore Nouve | TN | 4.8 | 3180 | 2289 | 3789 | 28.2 | N | 57.7% | 20.4% | Neigh. |
| Pièce | PI | 2.9 | 3046 | 2636 | 3784 | 27.8 | NE | 57.6% | 17.3% | Neigh. |
| Bertol Supérieur | BS | 2.6 | 3127 | 2913 | 3583 | 32.4 | SW | 9.2% | 14.3% | Nested |
| Vuibé | VU | 2.2 | 3036 | 2730 | 3722 | 24.7 | NE | 54.4% | 1.4% | Nested |
| Douves Blanches | DB | 1.5 | 3218 | 3097 | 3364 | 35.4 | W | 10.1% | 23.9% | Neigh. |

four subcatchments (Table 1): Bertol Supérieur (BS), Haut Glacier d'Arolla (HGDA), Mont Collon (MC) and Vuibé (VU). The BI catchment, with an area of 26.0 km$^2$, provides a good opportunity to test the transferability of hydrological parameters to nested catchments as there are three subcatchments that are also gauged upstream: BS (2.6 km$^2$ area), HGDA (13.2 km$^2$),

and VU (2.2 km$^2$). Remaining drainage to BI comes from the MC catchment or from points located between the BS, HGDA and VU gauges and the BI gauge (Fig. 1). Immediately to the north of the VU catchment is the Pièce catchment (PI), draining an area of 2.9 km$^2$; and the Tsijiore Nouve (TN) catchment with a drainage area of 4.8 km$^2$. On the other side of the valley, immediately to the north of the BS catchment is the Douves Blanches (DB) catchment with a drainage area of 1.5 km$^2$. These catchments allow us to test the transferability of hydrological parameters to neighboring catchments. The elevation of these

basins ranges from 2112 m a.s.l. (the elevation of the BI gauging station) to 3838 m a.s.l., the Grand Bouquetins peak, located in the Haut Glacier d'Arolla. At these elevations, it is extremely unusual to have such high-quality streamflow data for small, highly glacier-covered catchments.

The upper Arolla river basin presents a range of aspects (Fig. 1b), and its subcatchments have different general orientations (Fig. 1c). The glacial cover within the Arolla basin diminished over the years (Fig. 1d), from 66.5% in 1850 to 38.5% in 2016 for

the Bertol Inférieur catchment (GLAMOS, 2020). The geology of the study area consists mainly of metamorphic and igneous rocks, extensively covered with till and colluvial deposits (SwissTopo, 2024, Supplementary Fig. A3). The geomorphological characteristics of the subcatchments is generally similar. DB, BS, TN and the northern part of HGDA all present some rock glaciers (Lambiel et al., 2016), although their relative area is more significant for DB and BS.

Numerous studies have been carried out in the upper Arolla basin over the years, on topics ranging from glacier dynamics,

subglacial hydrology, sediment transport to hydrology (Sharp et al., 1993; Brock et al., 2000; Mair et al., 2002, 2003; Swift et al., 2002, 2005; Arnold, 2005; Pellicciotti et al., 2005; Dadic et al., 2010; Gabbud et al., 2015, 2016; Lane and Nienow, 2019), which makes this study area optimal for a technical study on hydrological parameter transferability.





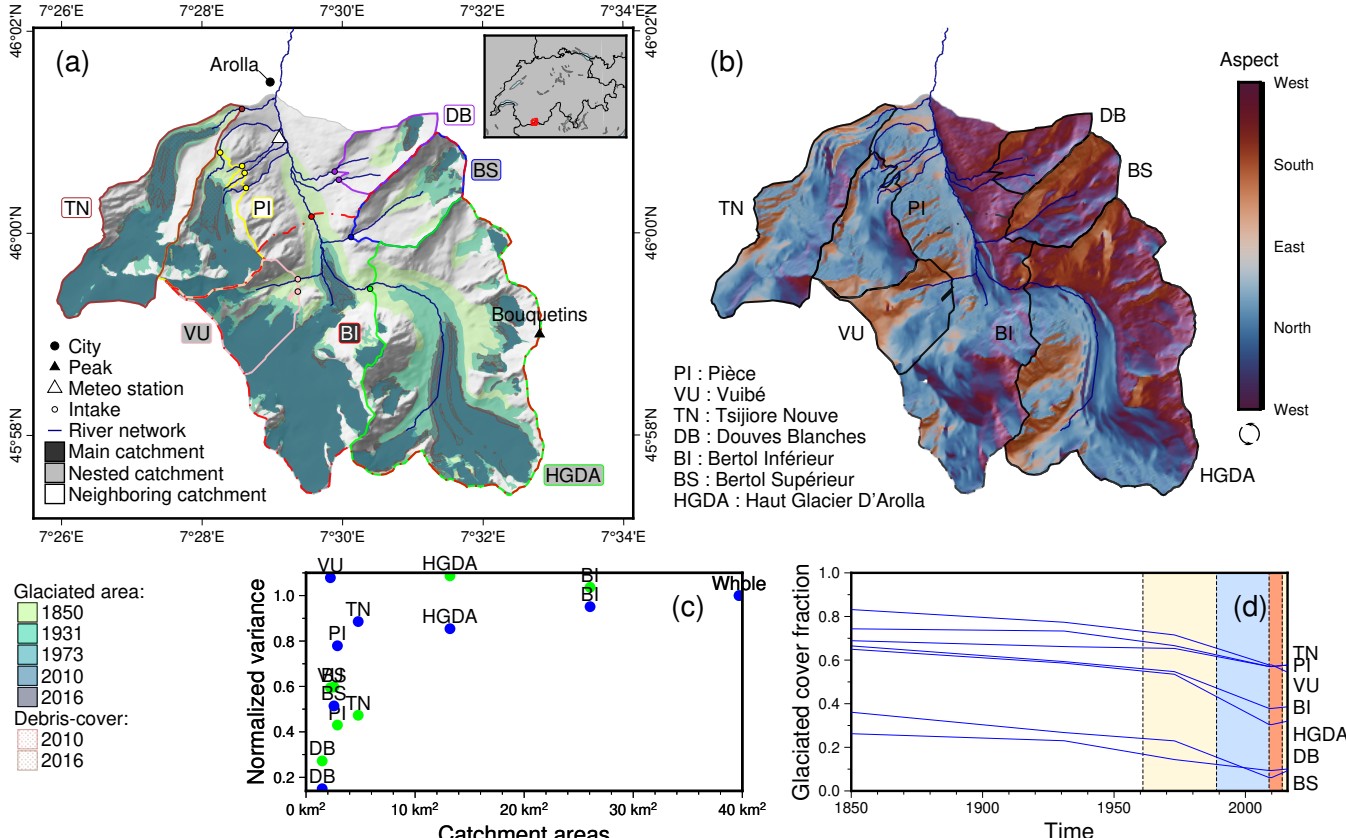

**Figure 1.** (a) Overview of the modeled catchments and subcatchments in the upper part of the Heremence (Arolla) catchment. Changes of the glacial cover through time are indicated in shades of yellow and green. The inset map shows the location of the study catchments in the Swiss Alps. (b) Aspect of the study area. (c) Aspect variogram derived from the aspect variance of each glacier (in blue) and each catchment (in green), normalized by the variance in the total glaciated area/total catchment (Whole), as done by Comola et al. (2015). (d) Glacial cover fraction through time, with the study period highlighted in orange, the available discharge period and meteorological data in blue and yellow, respectively. Topography is obtained from the SwissTopo DHM25 dataset (Swisstopo, accessed 2023) and glacier extents from the GLAMOS inventory.

## 2.2 Hydro-meteorological datasets

We use MeteoSwiss datasets for mean precipitation (MeteoSwiss, 2019a) and mean temperature available at 1 km resolution
for Switzerland (MeteoSwiss, 2019b).

Discharge data were provided by Grande Dixence SA (2024) at seven water intakes. In Switzerland, regulatory standards require hydroelectric power production companies to report water abstraction details to the authorities. In the upper Arolla river basin, discharge data was thus provided at a 15-minute resolution since 1971. Each basin features a calibrated water level recorder, initially utilizing a chart recorder and later upgraded to a pressure transducer with digital data logging. Water levels



are measured across a broad-crested weir, ensuring highly reliable discharge records ($\pm 0.01$ m$^3$/s for regulatory compliance). Discharge is measured in the intake. If the intake overflows, only part of the water is recorded. However, as any loss of water is a financial loss, the intake has been designed to capture capture practically all the discharge. Such overflows are therefore possible, but infrequent. The intakes defining the extent of each subcatchment are sometimes multiple, as with Douves Blanches and Vuibé, which both present two intakes, and Pièce, which presents four (See Fig. 1). Thus, the discharge is the sum of the

corresponding intakes.

With the exception of the Bertol Inférieur, discharge data were already preprocessed by Lane and Nienow (2019) to eliminate drawdown events linked to sediment removal during intake flushing. Since these drawdown periods typically last between 30 and 60 minutes, they can be visually recognized using the method outlined in the work of Lane et al. (2017). After the removal of data portions corresponding to such drawdowns, any missing data points were linearly interpolated. However, for the Vuibé

intake, data were unavailable from August 31 to December 31, 2011, due to intake maintenance work. In our study, we excluded this last period for Vuibé and applied the same preprocessing method to the Bertol Inférieur discharge time series, removing drawdown events and discarding associated time periods (in blue, Fig. 2). Furthermore, the water from the HGDA, BS and VU intakes is diverted and does not pass through the BI intake. We thus added the records of its upstream nested intakes to the BI discharge record (the actual measurements in BI is called BIrest, see Fig. 2). We propagated to BI the intake maintenance

work of VU and the drawdown removals of BIrest by discarding the affected time periods. We assume the time taken by the water to reach the BI intake from the upstream intakes to be negligible at the daily scale. Subsequently, the 15-minutes time step discharge datasets were summed up to daily time step datasets after the preprocessing.

Due to the confidentiality of the original discharge data, these datasets are shown here normalized by the highest observed discharge values (see figures 2, 4, 6, 8, 9, 11 and 14). The normalized dataset is called either "normalized discharge" when the

discharge was expressed in m$^3$ s$^{-1}$, and "specific normalized discharge" when the discharge was expressed in mm.

Glacier extents for the years 2010 and 2016 were obtained from the GLAMOS inventory (Fig. 1; GLAMOS, 2020; Linsbauer et al., 2021; Fischer et al., 2014).

This inventory specifies the debris cover extent for the year 2016. To obtain older debris cover trends, we used the algorithm developed by Shokory and Lane (2023), now available in an ArcGIS Pro toolbox, and computed the 2010 extents based on

Landsat Level 1 imagery (for details, see the Supplementary Material, Section C). We assumed the glacial cover of 2009 to be identical to 2010.

We derived the topography from the SwissTopo DHM25 dataset (Swisstopo, accessed 2023) available at 25 m resolution. From this topography, we automatically extracted the catchment areas, except for VU. VU requires manual correction of its South extent due to the presence of thick ice cover, which complicated the identification of the drainage divide (Fig. 6 of

Bezinge et al., 1989; Hurni, 2021).





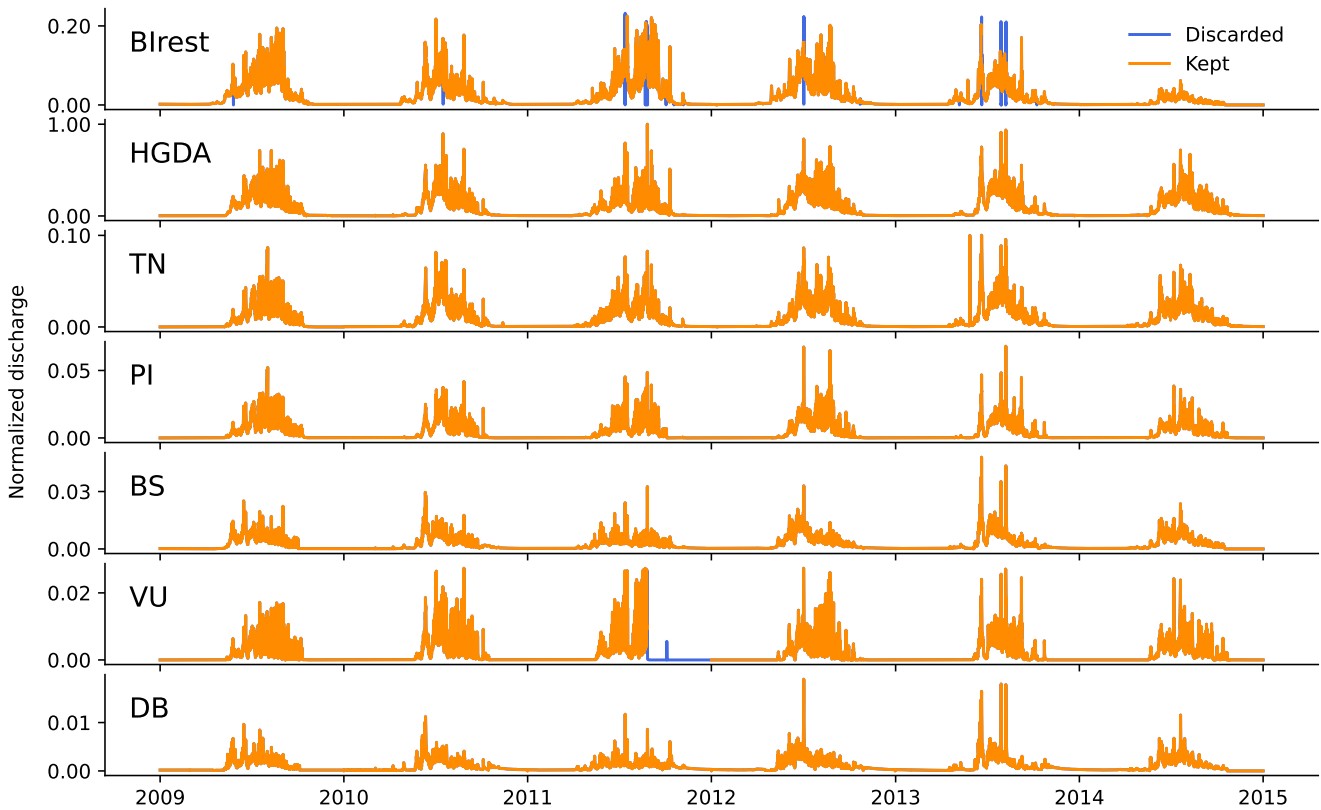

**Figure 2.** Observed discharge series for all subcatchments: Comparison of the discharge series kept for calibration in Hydrobricks (orange) with the discarded periods (blue). Discharge (unit: $m^3\ s^{-1}$) is normalized to the highest value.

## 3  Methods

### 3.1  Hydrological modeling with Hydrobricks

Hydrobricks comprises a hydrological modeling framework that implements the semi-lumped GSM-SOCONT model (Glacier and SnowMelt – SOil CONTribution; Schaefli et al., 2005) to simulate nivo-glacial hydrological regimes. The model consists of two main components: (a) the reservoir-based SOCONT model, which incorporates a linear reservoir method to account for slow storage contribution (emulation of subsurface ground water) and a non-linear reservoir approach to address quick runoff, and (b) the GSM model, which is specifically designed for glacier-covered catchments. The Hydrobricks framework is based on a C++ core integrated into a Python interface (Horton and Argentin, 2024), which allows for enhanced computing performances.

The model discretizes the catchment into hydrological response units (HRUs) by elevation, aspect and potential clear-sky direct solar radiation. The HRUs can have fractional land cover types, here 'glacier' for glacier-covered areas, and 'ground'





for non-glacier-covered areas. The distinction between debris-covered glacier areas ('glacier_debris') and debris-free glacier areas ('glacier_ice') can also be made (Shokory and Lane, 2023). The processes occurring within the same land cover type but in different HRUs are assigned identical parameters.

Following GSM-SOCONT's original structure, the model behavior differs between the glacier-covered area and the ice-free part. For the ice-free fractional part of a given HRU, surface and subsurface runoff components, along with baseflow from melt and rainfall, are computed per HRU and summed across all HRUs to build the non-glacier streamflow component at the outlet. For glacier-covered areas, the liquid water from melt and rainfall produced by each HRU is fed into to two lumped parallel linear reservoirs shared by all HRUs. For a detailed workflow, refer to Figure A1 and Schaefli et al. (2005).

The transition from rainfall to snowfall is defined in a fuzzy approach (Schaefli and Huss, 2011) between 0°C for the lower end ($T_{\text{s-r, min}}$) to 2°C for the upper end ($T_{\text{s-r, max}}$). Snow and ice start melting at a threshold melt temperature $T_T$ defined at 0°C, and ice can only melt when not covered anymore by snow.

We use the SPOTPY library (Houska et al., 2015) provided with Hydrobricks for parameter optimization with the Shuffled Complex Evolution algorithm of the University of Arizona (SCE-UA). The SCE-UA algorithm is designed to prevent remaining

stuck in local optima. We use it in combination with the Nash-Sutcliffe (NSE) (Nash and Sutcliffe, 1970) and Kling-Gupta efficiency (KGE) (Gupta et al., 2009) performance criteria to find the best combination of parameters (Table 2), after 10,000 simulations.

### 3.2    Hydrobricks developments

In the original version of GSM-SOCONT (Schaefli et al., 2005), precipitation type (snow or rain) is determined by a tem-
perature threshold and melt is calculated through a classic temperature-index melt model (TI). Two new melt models were implemented in Hydrobricks: the aspect temperature-index model (ATI) and the temperature-index model of Hock (HTI).

The aspect temperature-index model (ATI) is based on the discretization of the study area by aspect (North, South, East/West) and the use of a distinct degree day factor depending on aspect. A more complex model, the temperature-index melt model of Hock (HTI; Hock, 1999), links potential clear-sky direct solar radiation to melt rates (Eq. 2):

$$
\quad M_{\text{HTI}}(t) = \begin{cases} (m + r_j I_{\text{pot}})(T_a(t) - T_T) & : T_a(t) > T_T \quad \text{with } j \in \text{snow, ice} \\ 0 & : T_a(t) \leq T_T \end{cases} \tag{2}
$$

where $M_{\text{HTI}}$ is the melt rate (mm d$^{-1}$), $m$ is the melt factor common to both ice and snow (mm d$^{-1}$ °C$^{-1}$), $r_j$ is the radiation factor for ice or snow (mm d$^{-1}$ °C$^{-1}$ m$^2$ W$^{-1}$), $I_{pot}$ is the potential clear-sky direct solar radiation (W m$^{-2}$), $T_a$ is the air temperature and $T_T$ is the threshold melt temperature. Thus, while the ATI model represents a first attempt at handling spatial differences in melt rates, the HTI model has the benefit of directly taking into account irradiation, which should make it better
suited to reproduce melt rates in catchments influenced by aspect and cast shadows (e.g., Gabbi et al., 2014).

The HTI model requires computation of the potential clear-sky direct solar radiation $I_{pot}$, here implemented using the definition of Hock (1999, Eq. 3):





$$I_{\text{pot}} = I_0 \left( \frac{R_m}{R} \right)^2 \Psi_a^{\left( \frac{P}{P_0 \cos(Z)} \right)} \cos(\theta) \tag{3}$$

where $I_0$ is the solar constant (1368 W m$^{-2}$), $(R_m/R)^2$ is the Earth's orbit's eccentricity correction factor, composed of $R$
and $R_m$ the instantaneous and mean Sun-Earth distances, $\Psi_a$ is the mean atmospheric clear-sky transmissivity, $P$ and $P_0$ the
local and the mean sea-level atmospheric pressures, $Z$ the local zenith angle and $\theta$ the angle of incidence between the normal
to the grid slope and the solar beam. The potential direct solar radiation $I$ is set to 0 when a point is not directly hit by sunlight
(night time and cast shading brought by surrounding relief).

For the TI model, based on temperature only, the HRUs are evenly spaced elevation bands (Schaefli et al., 2005). For the
ATI and HTI models, the HRUs reflect the elevation variations as well as the aspect or the mean annual irradiation variations.
To avoid any HRU scaling influence on parameter transferability (Liang et al., 2004; Troy et al., 2008), we use a spacing of
40 m for elevation, 3 categories for aspect (North, South and East-West to group by degree of sun exposure) and a spacing of
65 W m$^{-2}$ for potential direct solar radiation for all catchments (Supplementary Fig. A2). Furthermore, in contrast to earlier
studies employing GSM-SOCONT/Hydrobricks, which relied on monitoring station data to derive meteorological lapse rates
across the different elevation bands, we needed to derive our meteorological input from gridded datasets. Our study therefore
adopts a distinct methodology, extracting meteorological input for each HRU directly from gridded datasets. This involves
quantifying how much the different cells in the gridded datasets contribute to each HRU. We do this by downscaling once the
grid of the meteorological data (1 km) to the DEM resolution (25 m) and computing the weights representing the contribution
of each data cell to each HRU based on their spatial coverage. These weights are then used to derive the mean values for each
HRU, for each daily time step. This allows direct use of future climate model outputs often provided as gridded datasets. The
evapotranspiration is then computed at the HRU level, from mean values of temperature, following the Hamon equation (Fig 3
Hamon, 1963).

In the case where we differentiate between debris-covered and debris-free glacier coverage, we also have to adapt the melt
models by introducing new melt parameters governing the ice melt. The TI model switches from a single parameter ($a_{\text{ice}}$) to
two parameters ($a_{\text{deb-free}}$ and $a_{\text{deb-cov}}$). The ATI model goes from three parameters ($a_{\text{ice,j}}$ with $j \in$ N, S, EW) to six parameters
($a_{\text{deb-free,j}}$ and $a_{\text{deb-cov,j}}$ with $j \in$ N, S, EW). The HTI model goes from two parameters ($m$ and $r_{\text{ice}}$) to three parameters ($m$,
$r_{\text{deb-free}}$ and $r_{\text{deb-cov}}$). The new melt parameters respect the same calibration ranges, but an inequality is added to constrain
lower melting rates of debris-covered ice.

Virtually all the snow in our study area melts every summer, making it unnecessary to model the firn separately.

## 3.3 Modeling approach

Our study of the transferability of nivo-glacial parameters from the TI, ATI and HTI melt models to nested subcatchments and
neighboring catchments (cf. section 2) can be divided into 4 steps:

1. Calibration runs: We calibrate the model on all subcatchments and neighboring catchments independently and compare
   them.





2. Transfer runs in nested catchments: We transfer the parameters calibrated on the main catchment, the Bertol Inférieur (BI), to its nested subcatchments (BS, HGDA and VI) to simulate their streamflow and compare the results to observed streamflow.

3. Transfer runs in neighboring catchments: As previous step but we transfer the parameters calibrated on the main catchment, the Bertol Inférieur (BI), to the neighboring catchments (TN, PI and DB).

4. Increased model complexity run: We repeat the three above points but calibrate and run the model with differentiation of debris-covered glacier and debris-free glacier areas.

For the first step of our study, we calibrate the model for all catchments individually using daily observed streamflow over the years 2009-2014. We chose this simulation period because the glacier-cover remains relatively stable (Fig 1d). For performance metric assessment, the first simulation year is discarded since it is assumed to initialize the system. These runs
are called "calibration runs" as the whole period is used for calibration, and no validation is carried out.

For the second and third steps of our study, we transfer the calibrated parameters from the calibration run of the main catchment to nested and neighboring subcatchments. As for the calibration runs, the first year is discarded. These runs do not include any calibration procedure and are called "transfer runs".

To analyze the effect of differentiating between bare ice melt and debris-covered glacier melt, we complete all of the above
steps twice: once assuming bare ice for the entire glacier area and once accounting for debris-cover.

## 3.4   Benchmark metrics

A key challenge in model calibration is to assess how good a calibrated model actually is since the commonly used metrics do not have an absolute meaning (Schaefli and Gupta, 2007). Here, we propose to assess how good the transferred runs are by assessing if they outperform bootstrapped time series. To do this, we perform block bootstrapping on the discharge series during
the evaluation period (2010-2014) with yearly block sizes. This process is repeated 100 times, ensuring that the streamflow data from a particular year is not used for that same year's prediction. We then compute the NSE and the KGE on each bootstrapped series and average them to obtain benchmark metrics. The benchmark NSE and KGE correspond to the prediction potential of the discharge dataset itself.

## 4   Results

### 4.1   Calibration runs

For the calibration runs without accounting for debris-cover, the NSE and KGE values are for all catchments better than those obtained from bootstrapping (Fig. 3; section 3.4), implying a consistent enhancement in streamflow modeling with Hydrobricks compared to a simple temporal transfer of observed data. This improvement is more pronounced in smaller catchments, as the values of the benchmark metrics decrease with decreasing catchment area, while the Hydrobrick simulation scores remain



**Table 2.** Parameters used in the simulations and their a priori range of values.

| Parameter (set) | Unit | Description | | Set value | Melt model |
|---|---|---|---|---|---|
| $T_{\text{s-r, min}}$ | °C | lower temperature threshold of the snow-rain fuzzy transition | | 0 | all |
| $T_{\text{s-r, max}}$ | °C | upper temperature threshold of the snow-rain fuzzy transition | | 2 | all |
| $T_T$ | °C | threshold melt temperature | | 0 | all |
| Parameter (calibrated) | Unit | Description | Condition | Range | Melt model |
| **MELT MODEL-DEPENDENT PARAMETERS** | | | | | |
| $a_{\text{ice}}$ or $a_{\text{deb-free}}$, $a_{\text{deb-cov}}$ | mm d$^{-1}$ °C$^{-1}$ | ice degree-day factor, independent (ice) or dependent on ice cover (debris-covered or debris-free) | $a_{\text{deb-cov}} < a_{\text{deb-free}}$ | 5 - 20 | TI |
| $a_{\text{snow}}$ | mm d$^{-1}$ °C$^{-1}$ | snow degree-day factor | $a_{\text{snow}} < a_{\text{ice}}$ or $a_{\text{deb-cov}}$ | 2 - 12 | TI |
| $a_{\text{ice,j}}$ or $a_{\text{deb-free,j}}$, $a_{\text{deb-cov,j}}$ with $j \in$ N, S, EW | mm d$^{-1}$ °C$^{-1}$ | ice degree-day factor, independent (ice) or dependent on ice cover (debris-covered or debris-free) and dependent on aspect (North, South, East/West) | $a_{\text{deb-cov,j}} < a_{\text{deb-free,j}}$ with $j \in$ N, S, EW | 5 - 20, 0 - 20 (North) | ATI |
| $a_{\text{snow,N}}$, $a_{\text{snow,S}}$, $a_{\text{snow,EW}}$ | mm d$^{-1}$ °C$^{-1}$ | snow degree-day factor, dependent on aspect (North, South, East/West) | $a_{\text{snow,j}} < a_{\text{ice,j}}$ or $a_{\text{deb-cov,j}}$ with $j \in$ N, S, EW | 2 - 12, 0 - 12 (North) | ATI |
| $m$ | mm d$^{-1}$ °C$^{-1}$ | melt factor | | 0 - 12 | HTI |
| $r_{\text{ice}}$ or $r_{\text{deb-free}}$, $r_{\text{deb-cov}}$ | mm d$^{-1}$ °C$^{-1}$ m$^2$ W$^{-1}$ | ice radiation factor, independent (ice) or dependent on ice cover (debris-covered or debris-free) | $r_{\text{deb-cov}} < r_{\text{deb-free}}$ | 0 - 1 | HTI |
| $r_{\text{snow}}$ | mm d$^{-1}$ °C$^{-1}$ m$^2$ W$^{-1}$ | snow radiation factor | $r_{\text{snow}} < r_{\text{ice}}$ | 0 - 1 | HTI |
| **RUNOFF TRANSFORMATION PARAMETERS** | | | | | |
| $k_{\text{ice}}$ | d$^{-1}$ | ice outflow coefficient | | | |
| $k_{\text{snow}}$ | d$^{-1}$ | snowpack outflow coefficient | $k_{\text{snow}} < k_{\text{ice}}$ | | |
| $k_{\text{quick}}$ | d$^{-1}$ | surface runoff outflow coefficient | | | |
| $A$ | mm | slow storage capacity | | 0 - 3000 | |
| $k_{\text{slow}_1}$ | d$^{-1}$ | slow storage outflow coefficient | $k_{\text{slow}_1} < k_{\text{quick}}$ | | |
| $k_{\text{slow}_2}$ | d$^{-1}$ | baseflow storage outflow coefficient | $k_{\text{slow}_2} < k_{\text{slow}_1}$ | | |
| $\rho_{\text{perc}}$ | mm d$^{-1}$ | slow storage percolation rate to the baseflow storage | | 0 - 10 | |

consistently high. This suggests that in smaller catchments, discharge signals present more robust yearly signatures than in bigger catchments, which can be effectively replicated using Hydrobricks, but not with bootstrapping. Given that, in general, KGE values tend to be higher than NSE values (Knoben et al., 2019), this trend is particularly apparent with the benchmark NSE and slightly less with the benchmark KGE. Thus, even though the achieved NSE values are often lower than those of KGE, the improvement they represent compared to the benchmark metric values is much bigger.

To illustrate how well the calibrated discharge simulations fit the observed discharge, Fig. 4 shows the corresponding hydrographs. The calibration runs based on NSE and KGE (Fig. 4) result in globally similar hydrographs for the TI and HTI melt



**Table 3.** Data used in the simulations, with corresponding source

| Dataset | Description | Acquisition period | Provider |
|---|---|---|---|
| Mean temperature | daily interval, 1 km resolution gridded dataset | since 1961 | MeteoSwiss |
| Mean precipitation | daily interval, 1 km resolution gridded dataset | since 1961 | MeteoSwiss |
| Discharge | 15-minute sampling time interval, measured at water intakes | since 1971 | Grande Dixence SA |
| Topographic data | 25 m resolution DEM (DHM25 dataset) | - | SwissTopo |
| Clean glacier extents | shapefiles of glacier extents | 1850, 1931, 1973, 2010, 2016 | GLAMOS inventory |
| Debris-covered glacier extents | shapefiles of debris-covered glacier extents | 2016 | GLAMOS inventory |
| Landsat imagery | Level 1 Landsat 7 imagery at 30 m resolution for debris-free ice mapping | 06/09/2009 | Landsat |

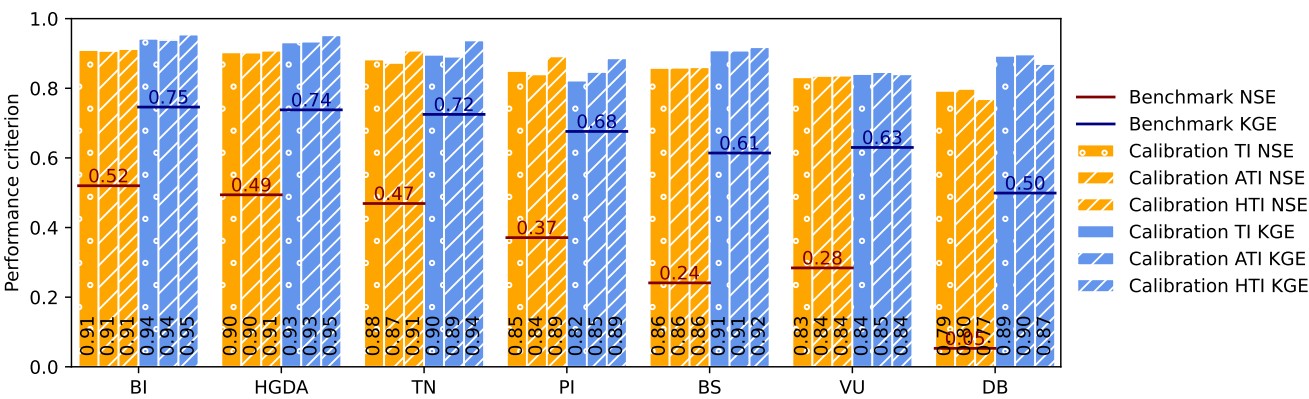

**Figure 3.** Comparison of the performance of the three melt models on the seven catchments for the period 2010-2014, quantified either by the Nash–Sutcliffe efficiency (NSE, orange bars) or by the Kling-Gupta efficiency (KGE, blue bars). For comparison, the benchmark NSE and KGE are computed and plotted as red and dark blue thresholds. The model is calibrated by running 10,000 times over the years 2009 - 2014, where 2009 is discarded for model initialization. Catchments are ordered by area, from BI (largest) to DB (smallest).

models, and for the ATI model (Fig. B1). While overall discharge dynamics are well simulated, discharge peaks are smoothed and thus not adequately reproduced. This is particularly evident in the case of the "spring event" occurring early- to mid-June. This first prominent peak during the melting season results from the melt of the supraglacial and hillslope snowpacks, that occurred due to an unusually strong foehn that blew on the 9th and 10th of June (Zbinden et al., 2010). This foehn event is partially recorded in the temperature records of the period (Fig. 14), which is why it could not be reproduced entirely.

Depending on the model used, the calibrated parameters are more or less similar between performance criteria and across catchments (Kruskal-Wallis test; Fig. 5). With the TI model, the calibrated parameter sets show very different values, depending on the catchment and the performance criterion used. With the ATI model, the degree-day factors and outflow coefficients







**Figure 4.** Observed and simulated hydrographs for all catchments for 2010 with the a) TI and b) HTI melt models. Observed discharge (black solid line) is compared to the calibration run using NSE (dotted orange) and KGE (dotted blue). Specific discharge (unit: mm) is normalized to the highest value.

obtained with KGE and NSE tend to converge to similar values, but these values still show a certain spread between the catchments. With the HTI model, the all parameter values are more consistent both between the two performance criteria and across the seven catchments. For example, the HGDA catchment shows high similarity with the BI catchment for both the snow radiation factor $r_{snow}$ and the ice radiation factor $r_{ice}$ in NSE calibration (non significant distribution change - ns; Fig. 5).

Thus, refining the representation of the melt process leads to increased spatial coherence of the melt parameters. The pa-
rameters showing no significant distribution changes between catchments could be assumed to be transferable between these







**Figure 5.** Calibrated ice and snow melt parameters for all simulated NSE and KGE runs for all catchments, with the three melt models and the two performance criteria. The parameter sets achieving the best NSE and KGE scores are plotted on top with a dot. Catchments are ordered by area, from BI (largest) to DB (smallest). The significance of the parameter distribution difference between BI and its neighboring and nested catchments is denoted as follows: *** for p < 0.001, ** for $p < 0.01$, * for $p < 0.05$, and ns for non-significant (Kruskal-Wallis test).

catchments without calibration. The HTI model can thus be assumed to be suitable to model the melt processes occurring in neighboring or nested catchments.

## 4.2 Transfer runs in nested catchments: spatial parameter transferability

To assess the spatial parameter transferability to nested subcatchments, we apply, for all melt models, the calibrated parameter
set obtained for the BI catchment to model the discharge of its nested subcatchments BS, HGDA and VI. The results for the TI and HTI model are shown in Fig. 6, and the ATI model in Fig. B2; the transfer runs match the observed discharge closely for the





**Figure 6.** Observed and simulated hydrographs of the BI catchment and its nested subcatchments for 2010 with a) TI and b) HTI melt models. Observed discharge (solid black line) is compared to the calibration runs and to the transfer runs with the calibrated parameters of BI: Shown are the results for NSE (orange) and KGE (blue); dotted lines show the calibration runs, solid lines show the transfer runs. For BI, the calibration and transfer runs are identical. Specific discharge (unit: mm) is normalized to the highest value.

TI and HTI models. For both models, the simulated discharges of the HGDA and BS catchments show slightly underestimated low flow periods and peaks. While this can be observed for HGDA throughout the summer, it is especially true for BS in the late summer. VU, on the contrary, shows slightly overestimated discharge in the low flows and the peaks, starting July, and overall, its discharge is best reproduced by the TI model. Close inspection reveals some differences between the transfer runs based on NSE versus KGE calibration, but no systematic differences.





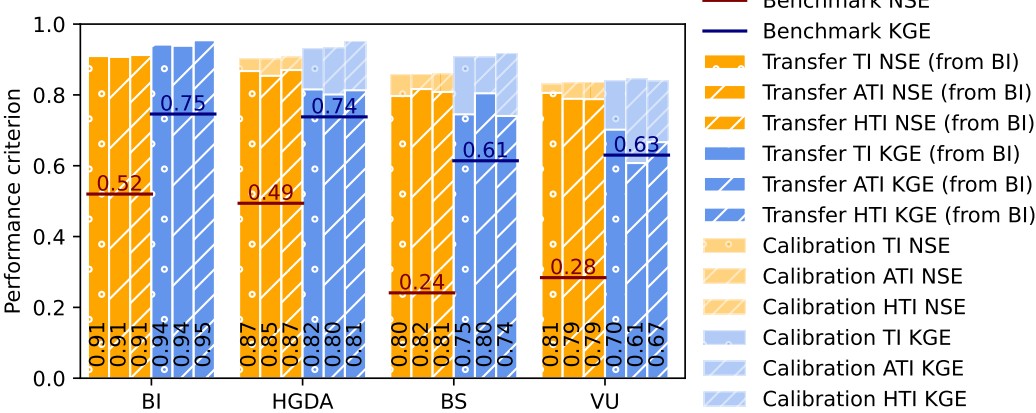

**Figure 7.** As Figure 3 but comparing calibration and transfer runs for the subcatchments of BI for the period 2010-2014. Shown are NSE values (orange) and KGE values (blue), along with the benchmark NSE value (red line) and KGE value (dark blue line) values. The performance values for the corresponding calibration run are shown in more transparent color.

As expected, the transfer runs reproduce the observed discharge less closely than the calibration runs for each of the catchments. The performance metric values of the transfer runs (Fig. 7) are, nevertheless, high compared to the benchmark values. Globally, the performance drop is bigger for KGE than for NSE but given the different sensitivities of the two metrics, they cannot be directly compared (see Section 4.1). Although all subcatchments and models experience a drop in KGE values, VU is the only catchment whose results drop below the KGE benchmark value with the ATI model. With the exception of the BS catchment, the best results are obtained with the TI and the HTI models.

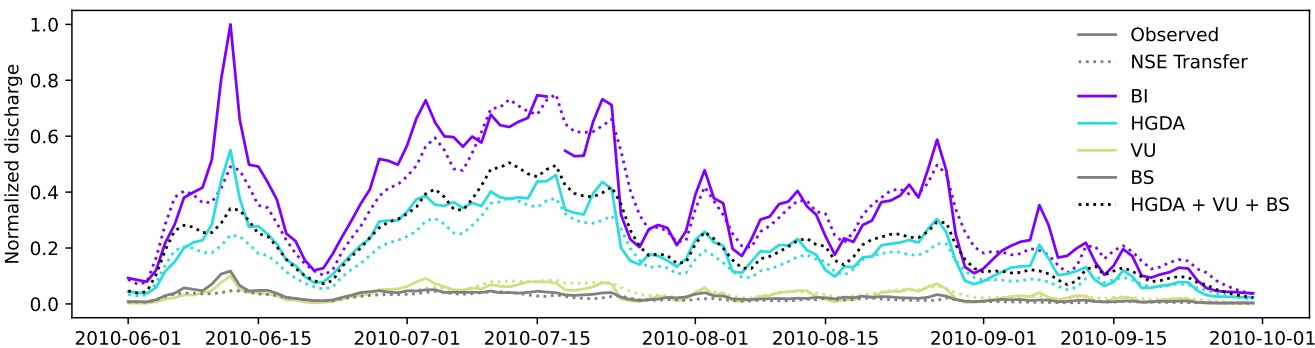

**Figure 8.** Observed and simulated hydrographs with the HTI model for the Bertol Inférieur and its subcatchments (VU, HGDA and BS) for the summer of 2010. For the subcatchments, the simulations (dotted lines) are the NSE-transfer runs, for the main catchment, BI, the dotted line corresponds to the calibration run. The solid lines are the observed hydrographs. The dotted black line shows the sum of the transfer runs of the three subcatchments. Discharge (unit: $m^3\ s^{-1}$) is normalized to the highest value.



We tested the conservativity of the model by checking whether the simulated discharges of the subcatchments (VU, HGDA and BS) were coherent across subcatchments and with the discharge of the main catchment (BI) (Fig. 8). This test is partial, as

the discharge generated by the Mont Collon (MC) area is not monitored, and thus the added discharges of VU, HGDA and BS do not account for the entirety of BI's discharge. We thus expect the sum of the three subcatchments' discharges to always be lower than the discharge of the main catchment BI, and to have a consistent overestimation/underestimation of the flow across the different subcatchments. We find that even for the low flow periods end of June and early September, which are slightly overestimated in BI, the stacked discharges (dotted black) stay below the BI discharge (dotted purple).

**4.3   Transfer runs in neighboring catchments: spatial parameter transferability**

The results of transferring the calibrated parameters of BI to neighboring catchments (Fig. 9t, B3) demonstrate for 2010 high accuracy in discharge simulation, both with the TI and HTI models, with minimal performance loss compared to calibration runs. The simulated discharge changes resulting from this forcing are more pronounced for the TI model than for the HTI model. Again, the discharge dynamics are relatively well reproduced but with a significant underestimation of the initial June

discharge peak for all catchments and the July ones for the TN and PI catchments. Interestingly, the July peaks are absent in the observed discharge of the DB catchment (as they are absent in the observed discharge of the BS catchment, see Figure 6). As seen in the nested forcing results (Figure 6), the NSE calibration run fits the discharge peak sometimes better than the KGE calibration run, such as in catchment DB in early July with the TI model.

We observe similar drops in NSE and KGE values when applying the BI parameters to the neighboring catchments as for the

transfer runs in the BI subcatchments (compare Figs. 7 and 10). Nevertheless, at the exception of the DB catchment with the ATI melt model, the performance decreases are less pronounced compared to the nested catchments. In all neighboring catchments, the simulations exhibit NSE and KGE values that surpass those obtained through bootstrapping. Thus, the catchments whose discharges are the least well simulated with BI's calibrated parameters are the BI's nested subcatchments: VU, BS and HGDA.

**4.4   Regionalization of the melt model**

Analyzing monthly discharge hydrographs (Fig. 11) can yield additional insights into KGE performance since this metric is by construction more sensitive to model biases than NSE, and such biases can become more apparent in monthly values compared to daily values. The monthly hydrographs (Fig. 11) clearly show the monthly discharge patterns that contribute to decreases in KGE, especially notable in 2012: In the VU catchment, discharge is overestimated, whereas in the HGDA and BS catchments, underestimations are observed.

Several factors could contribute to this discrepancy. Firstly, the meteorological forcing could be incorrect, especially the temperature, as it is highly variable in such alpine environments, difficult to observe, and available at a coarse resolution (1 km). Calibrated parameters are known to compensate for such input error effects (Bárdossy and Das, 2008) and transferred parameters might thus induce biases. Secondly, the delineation of some of the watersheds is uncertain, considering the uncertainty of water flow paths beneath glaciers. This might in particular be the case for VU, where Bezinge et al. (1989) suggest a potentially

smaller catchment area to the north.





 

**Figure 9.** Observed and simulated hydrographs of the BI catchment and its neighboring catchments for 2010 with a) TI and b) HTI melt models. Observed discharge (solid black line) is compared to the calibration run and to the transfer runs with the calibrated parameters of BI: Shown are the results for NSE (orange) and KGE (blue); dotted lines show the calibration runs, solid lines show the transfer runs. For BI, the calibration and transfer runs are identical. Specific discharge (unit: mm) is normalized to the highest value.

Thirdly, the melt model could be missing an important driving factor. Temperature-index based methods are known to yield good results in environments where melt is mainly driven by incoming longwave radiation and sensible heat flux (Ohmura, 2001), which is typically the case in Alpine catchments (Thibert et al., 2018).

In an attempt to investigate the third reason, we thus tried to attribute the performance decrease between calibration and transfer simulations to characteristics of the nested catchments (Fig. 12). The catchment area and mean catchment elevation do not show any obvious relations to performance decreases. However, the percentage of glacier debris cover and the mean






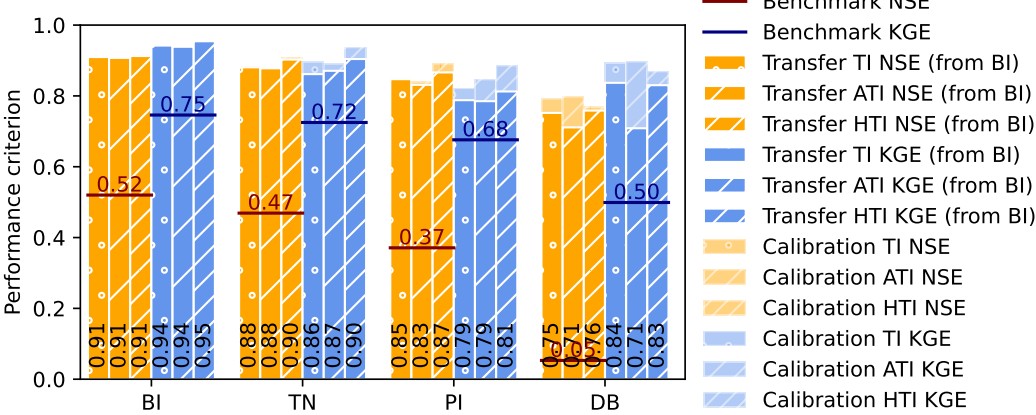

**Figure 10.** As Figure 7 but comparing calibration and transfer runs for the neighboring catchments of BI for the period 2010-2014. Shown are NSE values (orange) and KGE values (blue), along with the benchmark NSE value (red line) and KGE value (dark blue line) values. The performance values for the corresponding calibration runs are shown in more transparent color.

catchment and glacier slopes show more consistent relations. When the slope is steeper than in BI, the discharge is underesti-mated, whereas when the slope is shallower, the discharge is overestimated. In a similar way, when the debris coverage of the glacier is smaller than in BI, the discharge is underestimated. Accordingly, in a next step, we tested the transferability of model

versions that differentiate between debris-covered and debris-free glacier areas.

### 4.5   Increased model complexity run: accounting for debris-cover

With model versions that apply different melt and radiation factors to simulate melt from debris-covered and debris-free glacier areas, we obtain better model performances in the calibration phase (see Supplementary Material, Figure D). However, for the transfer runs, the performances are lower than for model versions that do not account for debris cover (Fig. 13b),

i.e. the transferability of the model parameters decreases. This is especially noticeable for the VU and TN catchments. The following reasons might explain this result: i) an overparameterization of the model (i.e. overfitting of local specificities), ii) a spatially inconsistent effect of debris cover on ice-melt or iii) difficulties to make the high number of parameters converge given the amount of reference information contained in observed discharge (which does not provide enough constraints on the parameters).

### 4.6   Physical meaningfulness of the transferred parameters

We showed that with the TI and HTI models it is possible to simulate the discharge of nested and neighboring catchments with parameters calibrated at the main local outlet (BI), albeit with a small decrease in performance. The ensuing question is to know whether or not these parameters can be used to infer conclusions about the physical processes and dynamics occurring



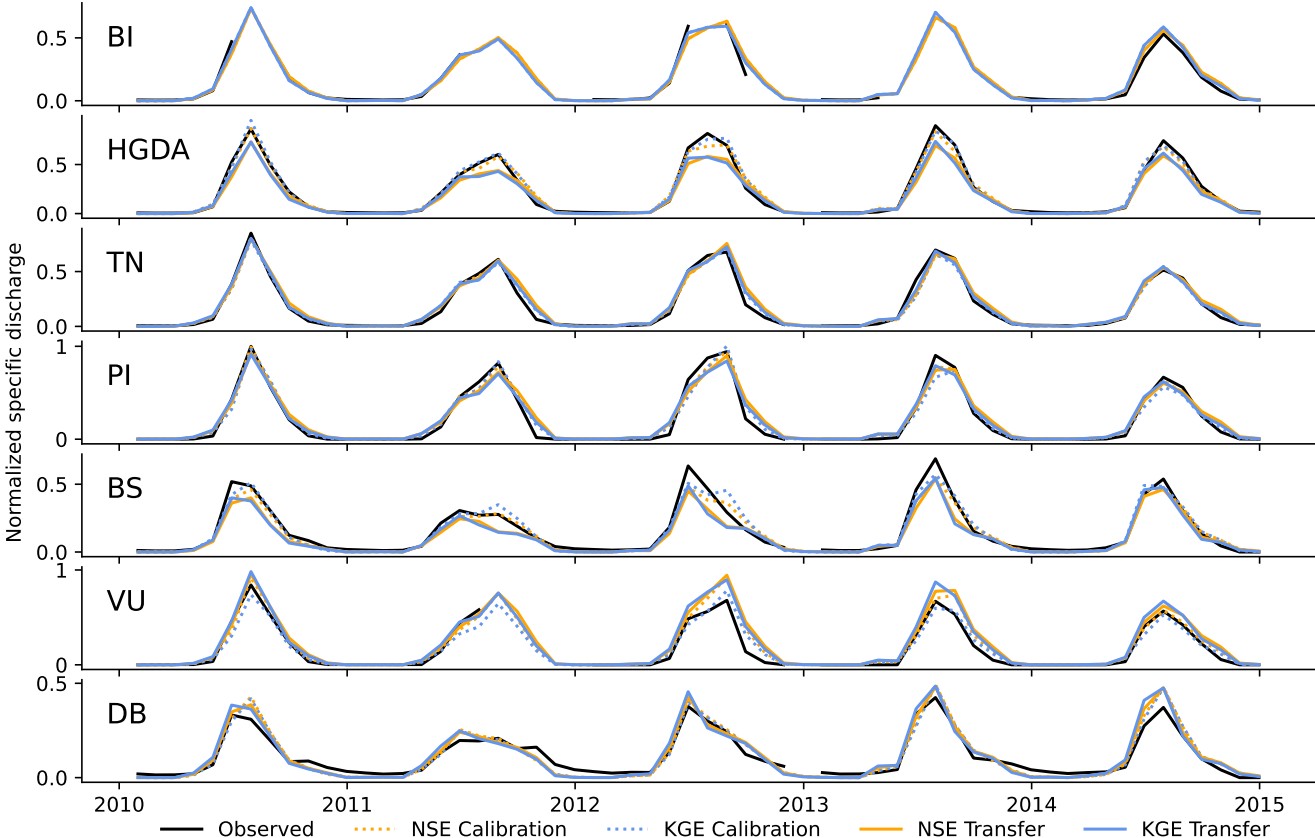

**Figure 11.** Observed and simulated monthly hydrographs for the HTI melt model on the seven catchments, calibrated or transferred with the calibrated parameter set of BI. Observed discharge (solid black line) is compared to the calibration run and to the transfer runs with the calibrated parameters of BI: shown are the results for NSE (orange) and KGE (blue); dotted lines show the calibration runs, solid lines show the transfer runs. For BI, the calibration and transfer runs are identical. Observed monthly discharges with missing values are not shown. Specific discharge (unit: mm) is normalized to the highest value.

in the neighboring and subcatchments. In our simulations, the studied catchments have very similar meteorological drivers, in
terms of precipitation (Fig. 14a) and temperature (Fig. 14b). The meteorological data is interpolated based on the ground-based observations from a few rather low elevation measuring stations, with the only station in our study area being that of Arolla at 2005 m (Fig. 1a) and the highest station in the bigger area being located at Col Grand St-Bernard, at 2472 m. Thus, the actual weather patterns may be more different between the studied subcatchments than what is given by the interpolated weather data. Indeed the studied catchment's discharge patterns show clear differences between DB and BS and the other catchments.
DB and BS show for all years on record a single melt-induced discharge peak in early summer and low discharge after this discharge peak. All the other catchments show the same discharge peak in June, followed by even higher discharges in the subsequent summer months (Fig. 14c).





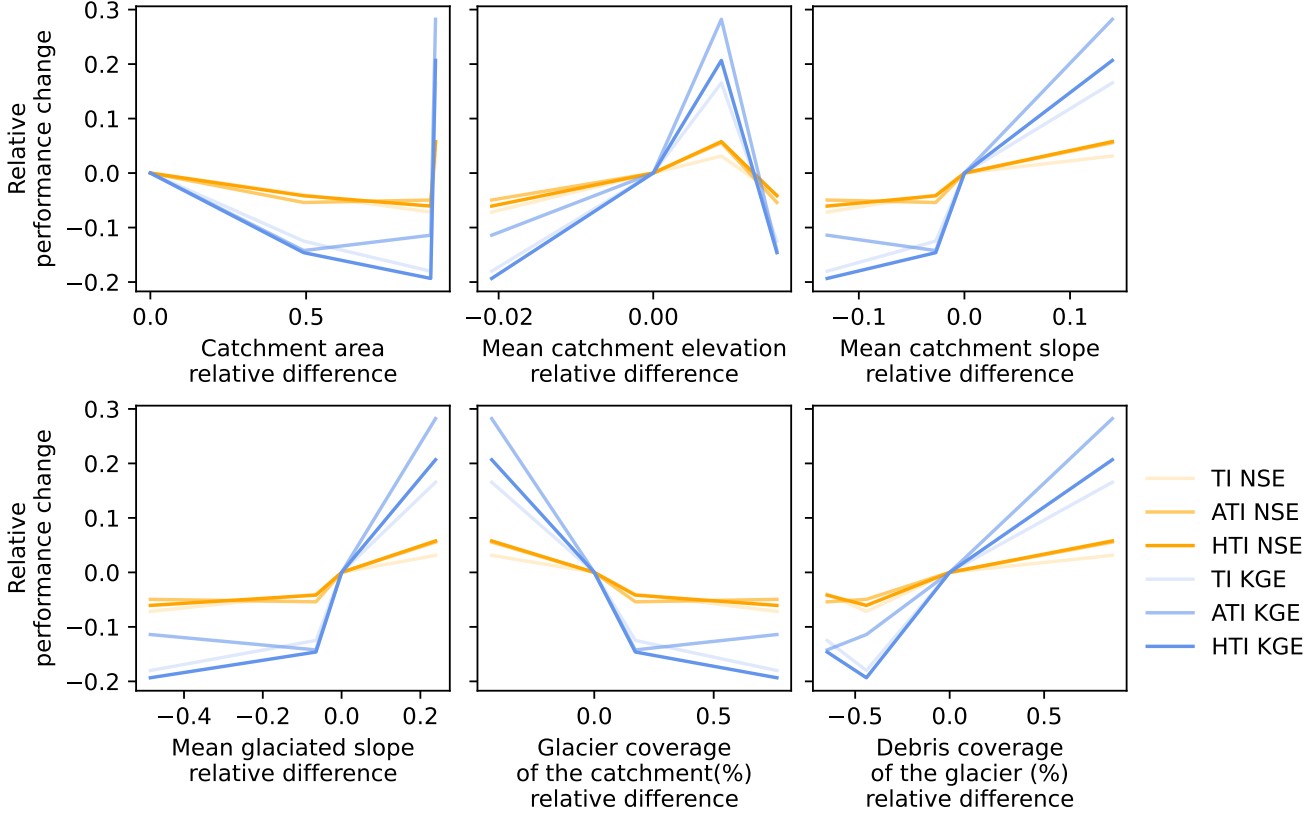

**Figure 12.** Comparison of selected physiographic characteristics of the nested catchments with the relative performance change between the calibration and transfer discharge simulations. The relative performance change is the relative drop in KGE and NSE performance criterion between the calibration run and the transfer run, multiplied by a coefficient reflecting the overestimation (1; VU) or the underestimation of simulated discharge (-1; HGDA and BS). The point showing no performance change is BI.

Based on the simulated snow water content, we find that the lower discharges exhibited by BS and DB in July-August cannot be explained completely by a snow exhaustion, as BS and DB still show more than 10 cm of mean simulated snow depth at the beginning of July. However, it can be explained by the intra-annual pattern of snow and ice melt - when snow melt slows due to depletion, glaciers become ice-free and ice melt begins. DB and BS show the lowest glacial coverage ($< 10.1\%$; Table 1), so when the snow has disappeared, very little ice melt ensues. In other catchments, ice cover is much greater ($> 32.0\%$), and in late July and early August, discharge is at its highest due to high rates of ice melt. We also note that TN is the only catchment for which the snow cover did not melt completely during summer 2010 (Fig. 14d).




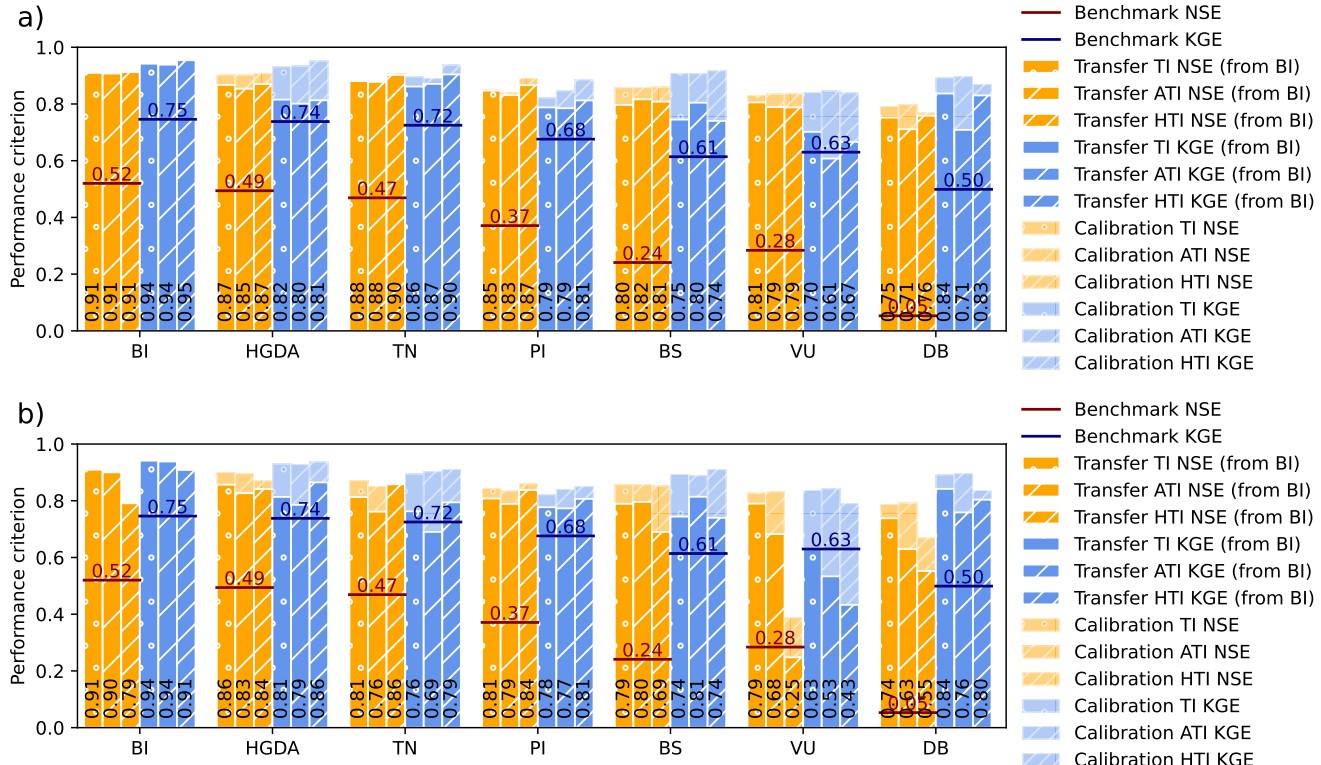

**Figure 13.** As Figure 3 but comparing calibration and transfer runs for all seven catchments, a) without or b) with debris-cover separation, for the period 2010-2014. Shown are NSE values (orange) and KGE values (blue), along with the benchmark NSE value (red line) and KGE value (dark blue line) values. The performance values for the corresponding calibration runs are shown in more transparent color.

## 5 Discussion

### 5.1 Discharge predictability: catchment size matters

We first discuss the influence of catchment size on discharge predictability, notably on our benchmark metrics. The benchmark metrics show that the predictability of discharge from past discharge signals alone is less high for small catchments than for bigger ones (Fig. 3). However, Hydrobricks shows similarly high model performances for small and larger catchments, highlighting the added value of a hydrological model in small catchments. This outcome can be directly explained by spatial relations. Large catchments exert a stronger averaging effect on spatio-temporal processes than small catchments. Indeed, the discharge in small catchments is driven by a localized and likely uniform meteorological patterns, while larger catchments draw from multiple local meteorological patterns, leading to a certain averaging. This complexity obscures the correlation between meteorology and discharge in larger catchments. Furthermore, the difference in catchment areas here closely linked to differences in the stream order, which results in a different balance between water travel times in unchanneled states (hillslopes,




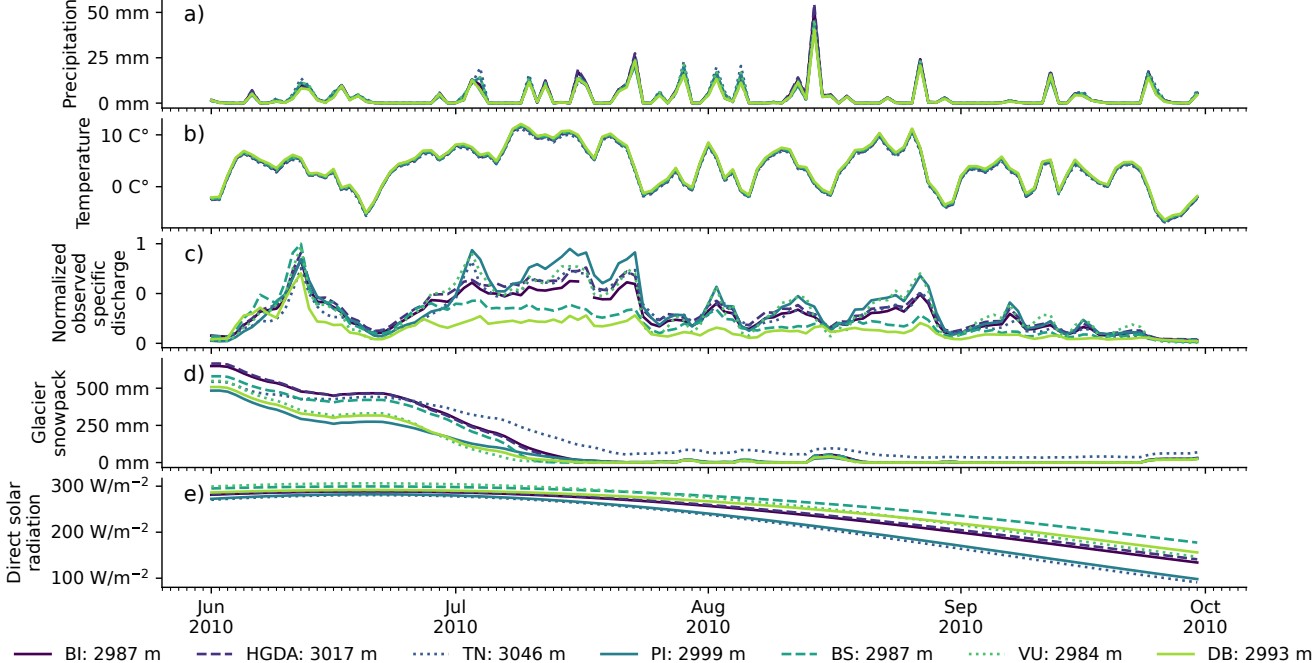

**Figure 14.** Patterns of a) precipitation, b) temperature, c) observed discharges, d) simulated glacier snowpack thickness and e) potential clear-sky direct solar radiation for the different catchments during the summer 2010. The values are mean values computed over the entire catchments from the RhiresD and TabsD MeteoSwiss datasets. The mean elevation of the catchment is indicated in the legend. Specific discharge (unit: mm) is normalized to the highest value.

surface runoff) and in channeled states (in-stream) (Rinaldo et al., 2006; Michelon et al., 2021). Longer in-stream flow paths lead hereby to a stronger dampening effect of hillslope- and glacier-scale runoff variability. Given the inherent year-to-year variability in meteorological patterns, and the close link between meteorology and discharge, it ensues that in small catchments, the discharge patterns from previous years are poor predictors of the current discharge. In contrast, even simple meteorology-
375 based hydrological models deliver much better results.

Additionally, we note that the NSE benchmark metric values tends to decrease much more strongly than the KGE benchmark metric values with decreasing catchment size (from 0.52 to 0.05 for NSE and from 0.75 to 0.50 for KGE; Fig. 3). This difference in decrease is explained by the bootstrapping approach used to produce the discharge data.

The NSE assesses the fit of one series to another solely based on the squared difference between the two time series. The
380 KGE, on the other hand, uses a linear combination of correlation between the two series, variability error (ratio of the standard deviations), and bias error (ratio of the means). Given the definition of KGE and NSE, the correlation term is linearly related to NSE, while the variability term and the bias term have a quadratic relation to NSE. As a result, the NSE is much more sensitive to changes in bias, changes in variability or shifted yearly patterns than the KGE (see Supplementary Material, F; Knoben et al., 2019). Thus, the benchmark KGE is a much harder criteria to meet for simulated discharges than the benchmark NSE.



## 5.2 Satisfactory hydrograph predictions across nested and neighboring catchments

As discussed previously, we expect the simulated hydrographs to outperform the benchmark NSE and match the benchmark KGE. When transferring the parameters calibrated for the largest subcatchment (BI) to model the discharge in all other nested (Fig. 7) and neighboring catchments (Fig. 10), we observe that despite exhibiting slightly inferior performance compared to the direct calibration, most catchments still show satisfactory results, even for the smallest ones. For all catchments, the transfer simulations with transferred parameters match both the benchmark KGE and NSE, at the exception of VU with the ATI model. The decreases in KGE for the HGDA, BS and VU observed in all melt models is not accompanied by a similar NSE decrease, which hints toward an amplitude change in the discharge signal (Supp. Mat. F). This amplitude change is produced by the underestimation (HGDA, BS) and overestimation (VU) in discharges that we observed in Figure 6.

The lowest NSE score obtained through a HTI transfer run is 0.76, and the biggest decrease with respect to the calibration run reaches 0.05. As a comparison, the best fits achieved by Parajka et al. (2005) for regionalization over Austrian catchments yielded a NSE decrease from 0.72 to 0.67, and globally, the mean and maximum NSE in European catchments reach 0.72 and 0.91, respectively (Guo et al., 2021). Furthermore, the discharges in nested catchments are consistent with each other (Fig. 8), as the sum of the nested discharges does not exceed the main catchment's discharge.

The performance decrease between calibration and transfer simulations in nested catchments could be attributed to the slope of the terrain and the debris coverage of the glacier (Fig. 12). When the catchment and glaciated slopes are steeper, the models tend to underestimate discharge, maybe because steeper slopes lead to faster runoff and higher discharge rates that are not fully captured by the model. Conversely, in shallower slopes, the models overestimate discharge, possibly due to slower runoff and more significant water retention. Additionally, lower debris coverage on glaciers leads to underestimated discharge, potentially because debris-cover, in our case, increases rather than decreases melt rates.

## 5.3 Enhanced parameter transferability through improved melt model

Melt rates per positive degree-day are sensitive to a number of characteristics that influence the surface energy balance, and which include elevation, direct solar radiation input, albedo, wind speed and seasonality (Hock, 2003; Ismail et al., 2023). These explain that for a given glacier, the degree-day factors of ice and snow are different, with ice, being less reflective than snow, melting more per positive degree-day. This variability of the link between positive air temperature and melt can also be found at a local-scale, inside snow and ice patches (Gabbud et al., 2015). This sensitivity of melt rates tends to limit the transferability of melt parameters from one catchment to a neighboring or nested catchment for the basic TI model.

By discretizing the study catchment by aspect and solar radiation and implementing the ATI and HTI models, we tested the influence of aspect and radiation on the melt model parameter transferability. According to the work of Comola et al. (2015), local-scale degree-day factors become stable (and therefore transferable) at scales at which the variability of local hillslopes' orientation does not further increase (less than 7 km$^2$, in their study). In this case study, we have five catchments that are small enough to be affected by their dominant aspect, but only two of them show low aspect variance also on their glaciated surfaces (DB and BS, Fig. 1c). However, all TI and ATI calibrated melt and radiation factors are highly inconsistent across catchments,



whereas we find some parameter overlap between catchments for the HTI model (Fig. 5). Similar to the work of Comola et al. (2015), we find that taking into account solar radiation patterns does not fully explain the hydrological response variability at smaller catchment scales. Indeed, BS and HGDA tend to have higher melt parameters when calibrated alone (Fig. 5), and produce slightly underestimated discharge when transferred with BI's parameters (Fig. 6), responsible for their lower KGE values (Fig. 7a). The transfer results obtained from the HTI model do not demonstrate improvements in terms of simulated hydrographs compared to the TI model, suggesting that the radiation as computed by Hock (1999) may not be enough to explain the KGE differences. We elaborate on Comola et al. (2015) to conclude that the obtained hydrographs are still very good fits for smaller, nested catchments, and that parameter transferability to catchments below 7 km$^2$ in the HTI model is a reasonable approximation.

These differences could also be due to differences in ice albedo. Indeed, the glaciers in the studied catchments are not equal in terms of debris cover (Fig. 1). We thus tried to take the debris into account, but failed to obtain better results in transfer run (Fig. 13b), which hint towards a non-consistent behavior of the debris cover, or an overparametrization of the model. Both explanations are possible as debris cover is known to either shield or amplify melting (Gabbud et al., 2015), and the contribution of two processes so close as debris-free ice melt and debris-covered ice melt would be hard to constrain from discharge data only (Pokhrel et al., 2008). Thus, debris cover related parameters are less transferable than the global ice parameters.

## 6 Conclusions

In this study, we tested the spatial transferability of melt models incorporating progressively more spatial information: a classical temperature-index melt model (TI), a temperature-index melt model based on the aspect (ATI) and the temperature-index melt model of Hock (HTI). To do so, we calibrated each melt model over seven different catchments, then transferred the calibrated parameters of the main catchment to the three nested catchments, and the three neighboring catchments.

The results show that for high alpine catchments, it is possible to spatially transfer relatively simple semi-lumped glacio-hydrological models. We have demonstrated that our semi-lumped model (Hydrobricks), can successfully simulate discharge at several upstream points of the catchment after calibration to a single downstream observed discharge time series. This makes multi-point discharge simulation possible.

Although the best results in terms of transferability are achieved with the TI and HTI models, the highest consistency between parameters is achieved with the HTI model. This better convergence of parameters is witnessed both between the two performance metrics, as is also the case for the ATI model, but also between the seven catchments. The inclusion of debris cover on glaciers does not produce better results, and leads to model overparameterization. The NSE metric gives best calibration results when focusing on attenuation of discharge trend offsets, but the benchmark KGE shows to be a harder, thus more significant, criteria to meet, and reproduces better the observed peaks. Thus, we find that the best framework to transfer parameters calibrated in the biggest local catchment to subcatchments and neighboring ones is by using the HTI model, without debris cover.





450      Our simulations highlighted the possible influence of catchment and glaciated slope, as well as debris-cover percentage on the overestimation and underestimation of discharge in transfer runs. Since the inclusion of debris-cover led to overparametrization, future research should focus on the integration of these characteristics in more spatially-informed ways.

*Code availability.* The software used to carry out this study is available at Horton and Argentin (2024).





## Appendix A: Discretization and lithology of the study area

**Figure A1.** Illustration of the Hydrobricks model workflow used in this study. The glacier-covered part illustrates the behavior of both the bare ice and debris-covered glaciers. Orange reservoirs are distributed over all elevation bands and red reservoirs are lumped over the catchment. Figure taken from Shokory et al. (2023).



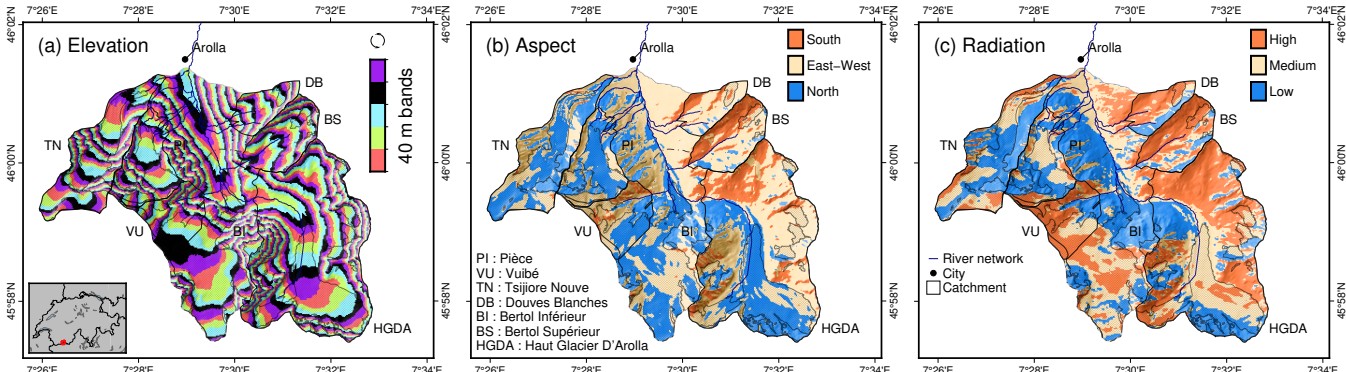

**Figure A2.** Hydrobricks' hydrological units for the whole catchment, discretized (a) according to elevation to use in the classic temperature-index (TI) model, (b) according to aspect to use in combination with elevation discretization in the aspect temperature-index (ATI) model (c) according to mean annual potential clear-sky direct solar radiation with cast shadows to use in combination with elevation in the Hock temperature-index (HTI) model.

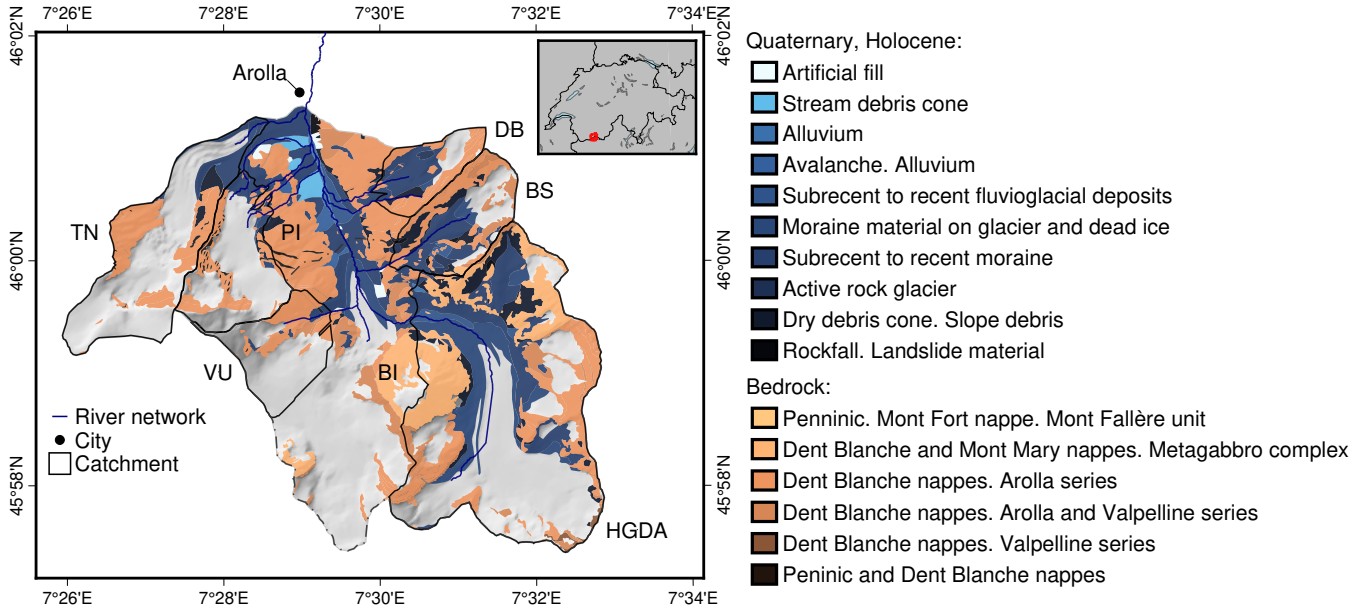

**Figure A3.** Geological cover of the study area, and of the different subcatchments, extracted from the GeoCover V2 product (SwissTopo, 2024).



| Catchment | Minimum elevation (m) | Maximum elevation (m) | Mean elevation (m) | Standard deviation elevation (m) | Mean catch. slope |
|---|---|---|---|---|---|
| Whole | 2183 | 3789 | 3085 | 289 | 29.3 |
| BI | 2183 | 3722 | 3063 | 229 | 28.7 |
| HGDA | 2582 | 3677 | 3014 | 191 | 29.5 |
| TN | 2289 | 3789 | 3180 | 443 | 28.2 |
| PI | 2636 | 3784 | 3046 | 266 | 27.8 |
| BS | 2913 | 3583 | 3127 | 117 | 32.4 |
| VU | 2730 | 3722 | 3036 | 152 | 24.7 |
| DB | 3097 | 3364 | 3218 | 58 | 35.4 |

**Table A1.** Basic statistics on the topography of the glaciated areas (2016) of each catchment.

| Catchment | Mean (glaciated) | SD (glaciated) | Mean (catchment) | SD (catchment) |
|---|---|---|---|---|
| Whole | 356.3 | 60.6 | 341.1 | 93.7 |
| BI | 347.9 | 59.1 | 323.4 | 95.4 |
| HGDA | 343.1 | 56.0 | 304.3 | 97.7 |
| TN | 12.6 | 57.1 | 15.4 | 64.5 |
| PI | 24.7 | 53.5 | 28.7 | 61.5 |
| BS | 238.9 | 43.5 | 237.7 | 72.4 |
| VU | 38.4 | 63.0 | 64.9 | 72.2 |
| DB | 249.7 | 23.4 | 259.2 | 48.9 |

**Table A2.** Circular means and standard deviations of the aspect over the glaciated areas (2016) and total areas of each catchment, computed with the zonal statistics of ArcGIS. SD: Standard deviation.

| Catchment | Debris cover area (km$^2$) | Glacier area (km$^2$) | Debris coverage percentage |
|---|---|---|---|
| BI | 1.00 | 10.04 | 9.9% |
| HGDA | 0.69 | 4.22 | 16.3% |
| TN | 0.56 | 2.76 | 20.4% |
| PI | 0.29 | 1.66 | 17.3% |
| BS | 0.03 | 0.24 | 14.3% |
| VU | 0.02 | 1.21 | 1.4% |
| DB | 0.04 | 0.15 | 23.9% |

**Table A3.** Debris cover areas, glaciated areas and debris cover percentage for each catchment for the year 2016.



455    **Appendix B:  Additional results with the ATI melt model**



**Figure B1.** Same as figure 4, but with the a) TI and b) ATI melt models.





**Figure B2.** Same as figure 6, but with a) TI and b) ATI melt models.



**Figure B3.** Same as figure 9, but with a) TI and b) ATI melt models.



## Appendix C: Debris cover mapping

The GLAMOS dataset offers both debris-free ice extent and glacier extent for 2016, but only glacier extent for 2010 and previous years (Linsbauer et al., 2021; Fischer et al., 2014). To obtain the debris-free ice extent trend since 2010, we relied on the debris-free ice detection algorithm from (Shokory and Lane, 2023), now available under ArcGIS Pro. We applied it to compute the corresponding debris-free ice extents for the 2010 GLAMOS dataset, thus allowing us to infer the debris cover evolution from 2010 to 2016.

Given the suboptimal conditions of Landsat 7 images in 2010 for mapping, we opted for an image from 2009. Two images, dated 06/09/2009 and 22/09/2009, displayed minimal cloud cover and limited snow patches. Between the two, the 06/09/2009 image displayed the smallest swath gaps. We corrected the Landsat 7 Level 1 near infrared (NIR)-B4 and shortwave Infrared (SWIR)-B5 bands, both available at 30m resolution, for top of atmosphere reflectance with solar angle correction. To do so, we applied the radiometric rescaling coefficients given in the associated metadata files provided with the Landsat Level-1 NIR and SWIR bands. We then applied the methodology of Shokory and Lane (2023) that uses the condition $\frac{NIR}{SWIR} \geq t$, with NIR representing the Near Infrared band, SWIR the Shortwave Infrared band, and $t$ denoting the threshold condition for debris-free ice delineation. We tested incremental thresholds with steps of 0.05 between 1.00 and 3.00 and determined that a threshold value $t$ of 2.00 provided the best results in the transition areas between debris-free ice and debris-covered ice (in brown, Fig. C1). We nonetheless had to manually correct for the influence of the swath gaps (in red, Fig. C1).

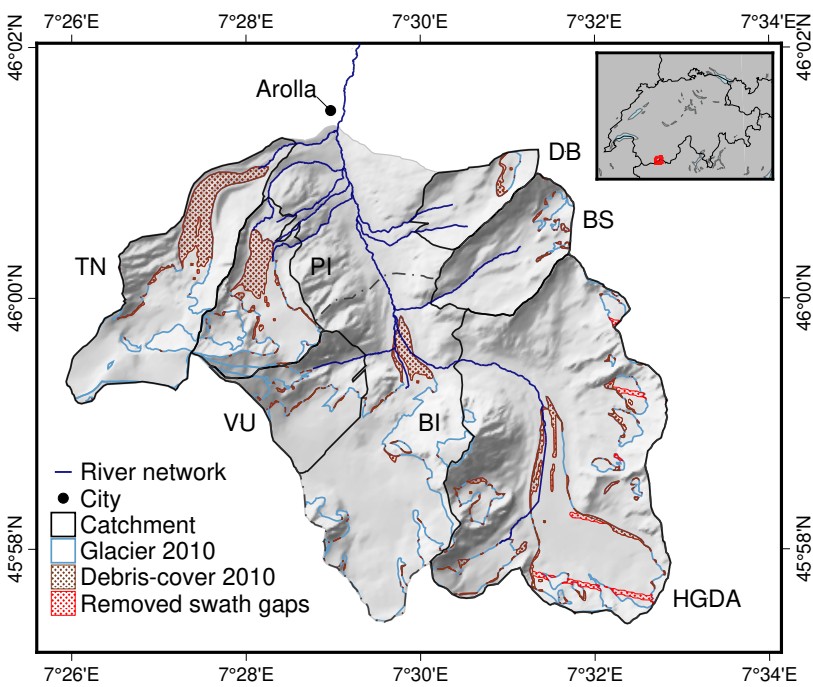

**Figure C1.** The mapped 2010 debris cover extent is indicated in brown, and the GLAMOS 2010 glacier extent in blue (Fischer et al., 2014). The manually removed debris linked to the swath gaps are indicated in red.





## Appendix D: Additional results with debris cover

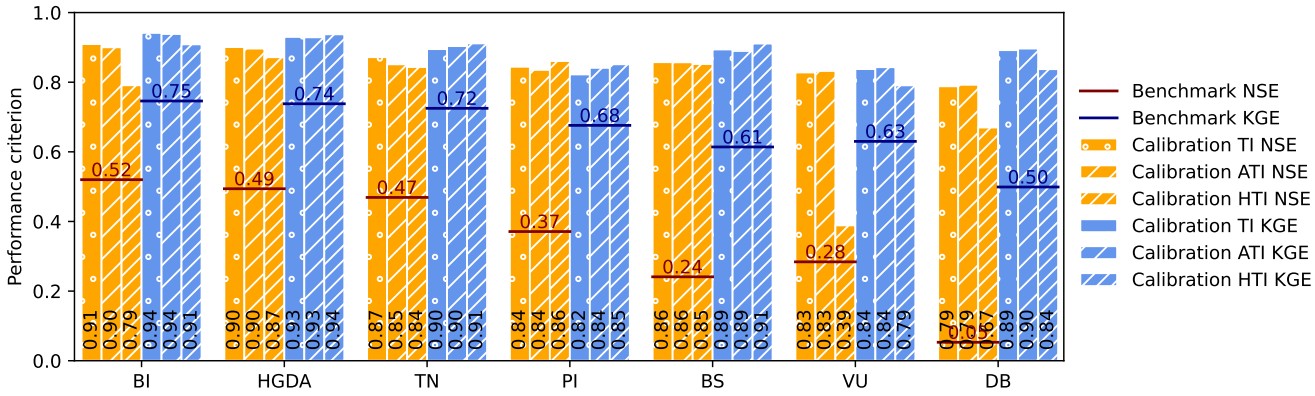

**Figure D1.** Comparison of the performance of the three melt models on the seven catchments, quantified either by the Nash–Sutcliffe efficiency (NSE, orange bars) or by the Kling-Gupta efficiency (KGE, blue bars) performance criteria of observed and simulated discharges for the period 2009-2014. For comparison, the benchmark NSE and KGE are computed and plotted as red and dark blue thresholds, respectively. The simulations are run 10000 times over the years 2009 - 2014, with 2009 the calibration year. Catchments are ordered by area, from BI (largest) to DB (smallest). All performance criteria are computed on the 2010-2014 time period.





**Figure D2.** Obtained ice and snow parameters for the best 5% NSE and KGE scores for all catchments, with the three melt models and the two performance criteria.





**Figure D3.** Obtained ground parameters for the best 5% NSE and KGE scores for all catchments, with the three melt models and the two performance criteria. The parameter set values achieving the best NSE and KGE scores are plotted on top with a dot. Catchments are ordered by area, from BI (largest) to DB (smallest).





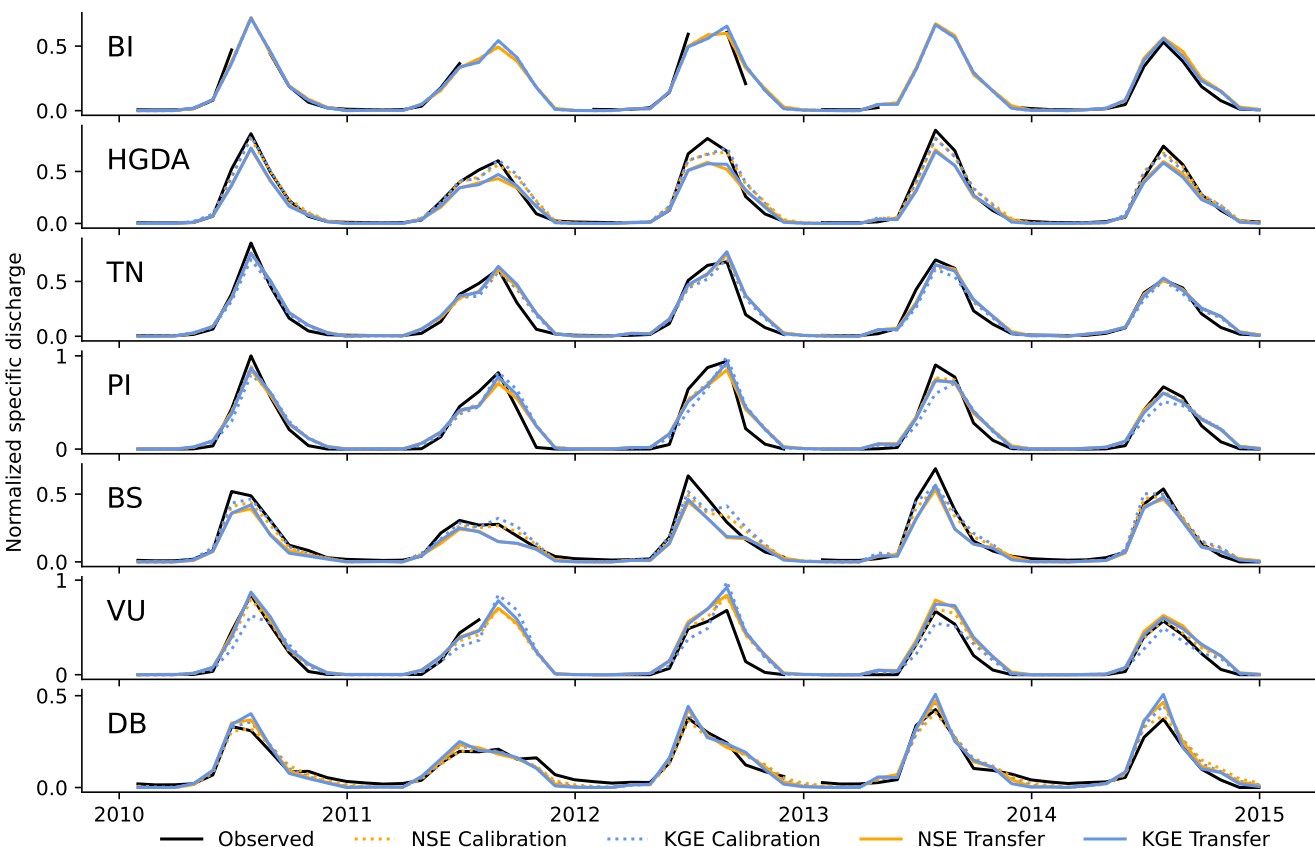

**Figure D4.** Monthly water hydrographs for the TI melt model on the seven catchments, calibrated or transferred through the application of the parameter set of BI. The original observed dataset (black) is compared to the calibration run using the NSE (dotted orange) and the KGE (dotted blue), and with the transfer run with the calibrated parameters found in the BI catchment with the NSE (orange) and the KGE (blue). Observed monthly yields with missing discharge values are not computed. Specific discharge (unit: mm) is normalized to the highest value.



## Appendix E: Patterns of water fluxes and retention

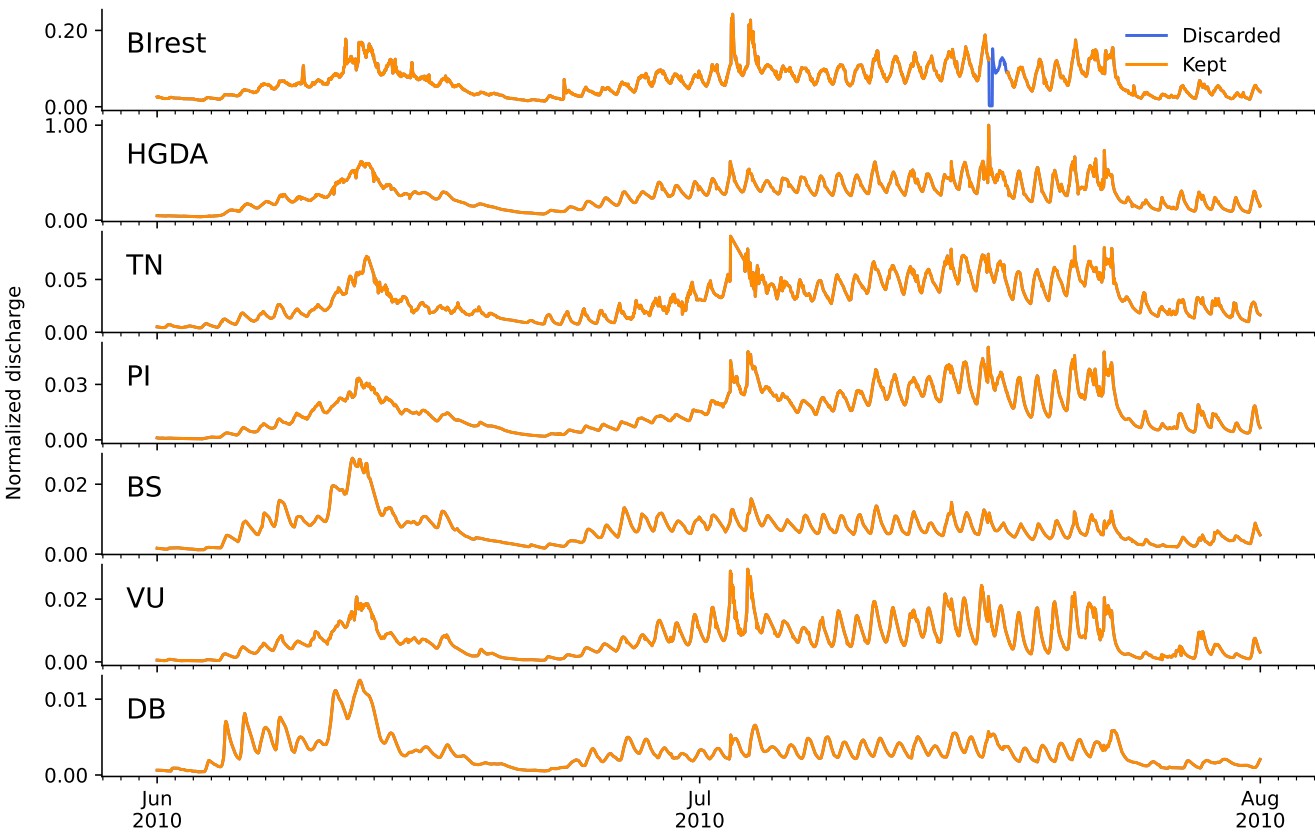

**Figure E1.** Comparison of the discharge series kept for calibration in Hydrobricks (orange) with the discarded periods (blue), over the summer 2010. Analysis according to Swift et al. (2005) leads to the interpretation that glacial snowpack was removed from mid-June on, allowing diurnal discharge patterns to take on a peaked shape. Discharge (unit: $m^3\ s^{-1}$) is normalized to the highest value.



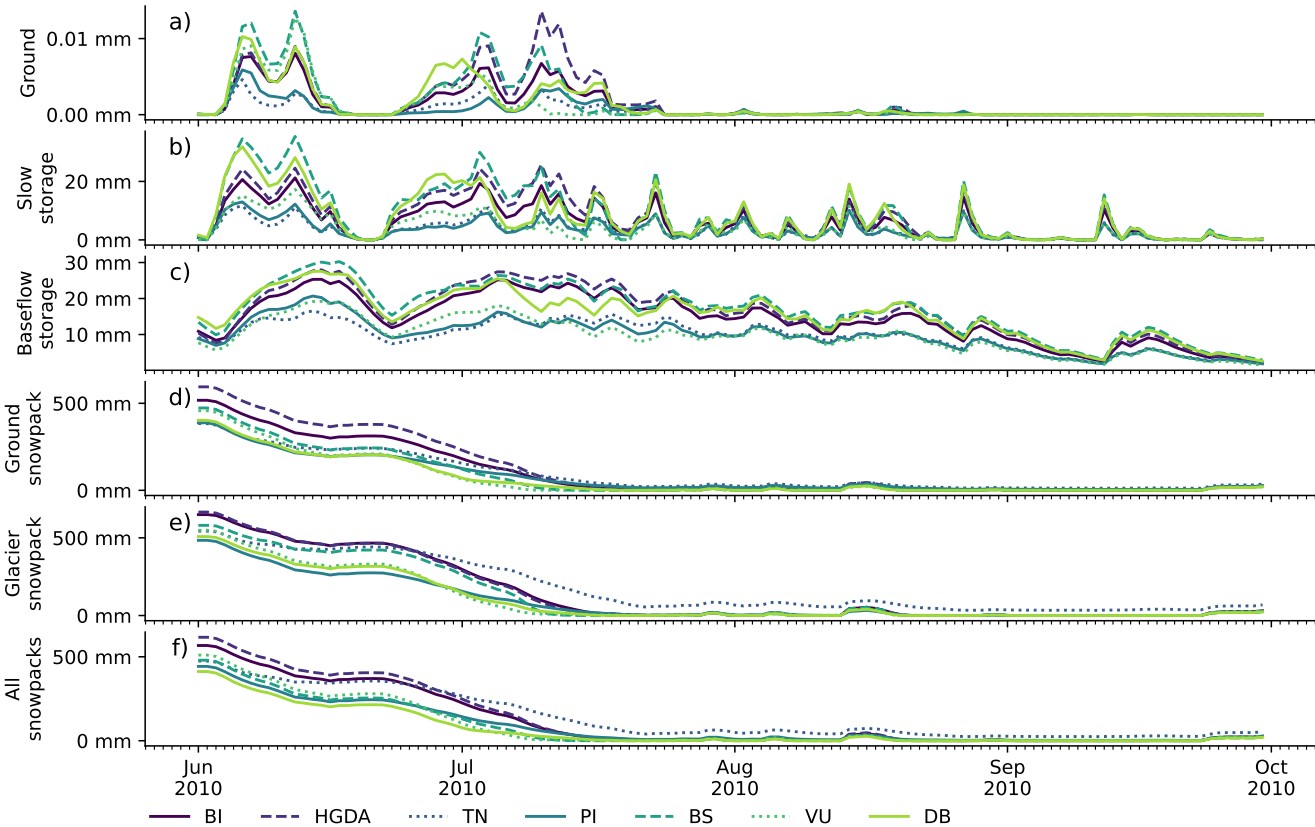

**Figure E2.** Water content heights in the a) ground, b) slow storage and c) baseflow reservoir, and snow water equivalent on the d) ground, e) glacier and f) ground and glacier during the summer 2010. Water heights are computed on their respective areas.



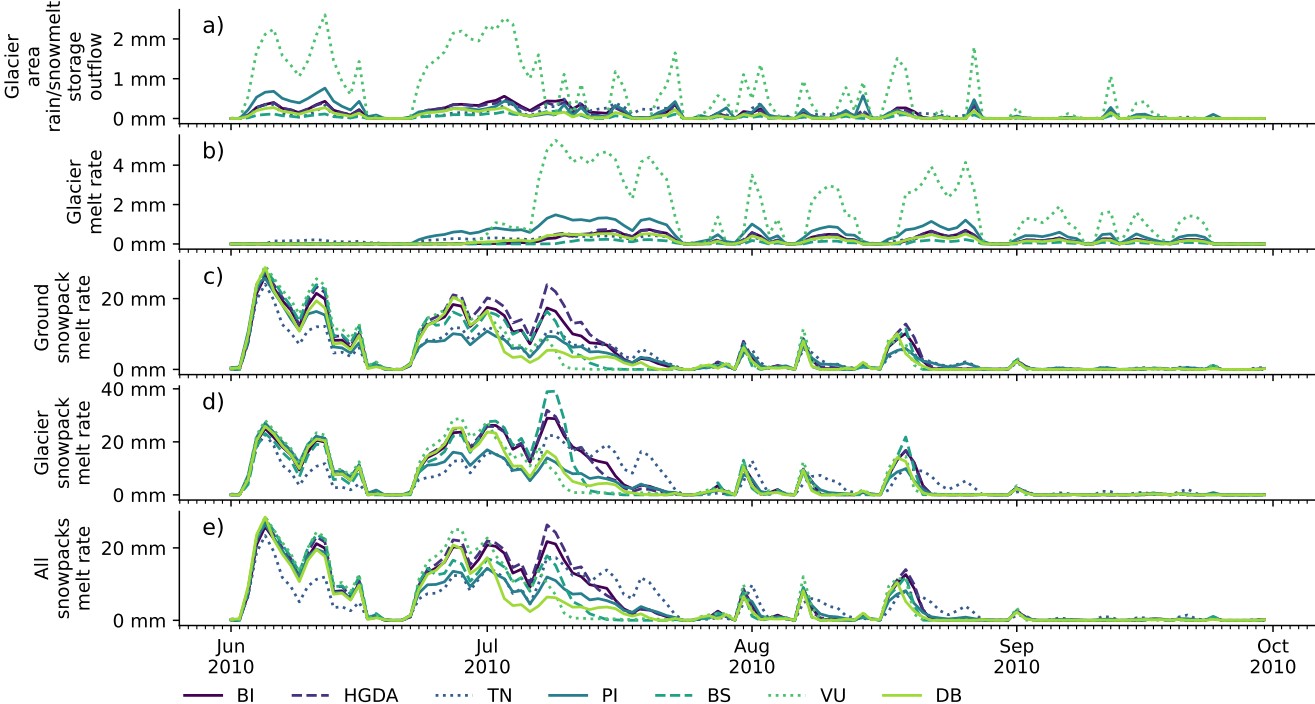

**Figure E3.** Water fluxes rates due to a) outflow of the glacier area rain/snowmelt storage, b) glacier melt rate, c) ground snowpack melt rate, d) glacier snowpack melt rate and e) global snow melt rate during the summer 2010. Melt rates are computed on their respective areas.



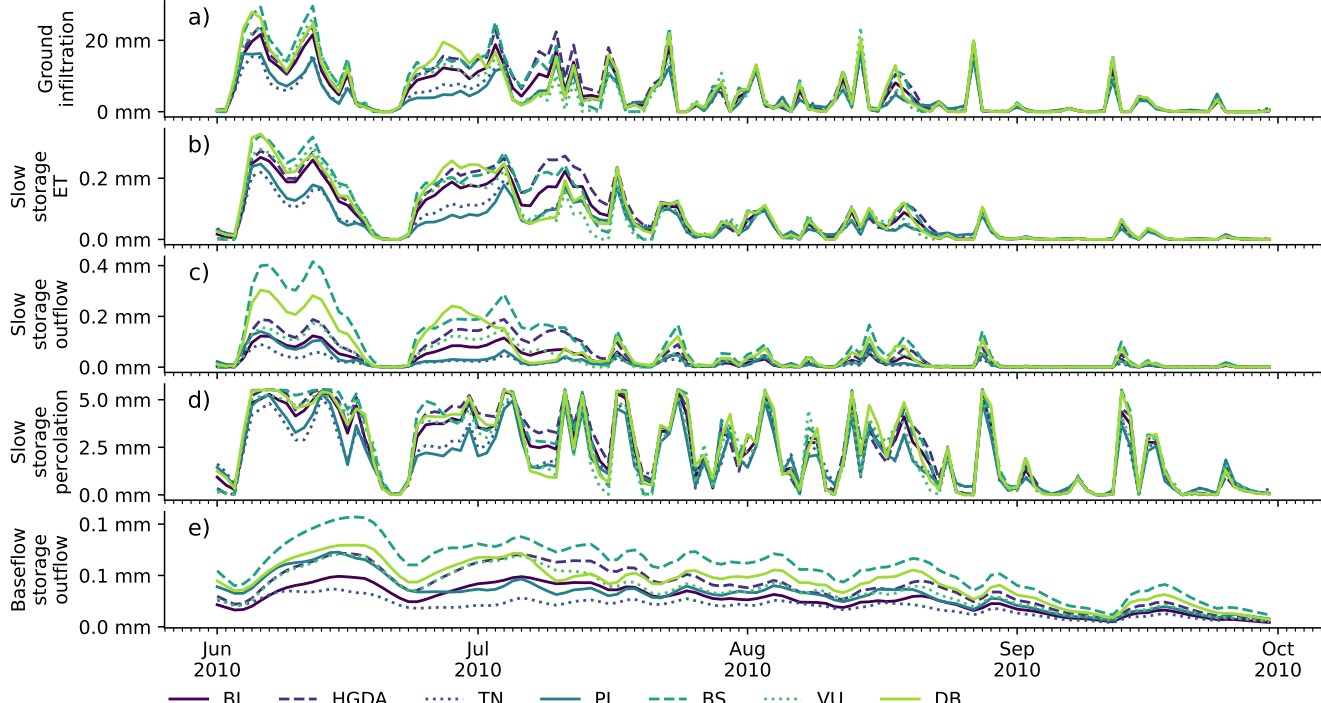

**Figure E4.** Water fluxes rates due to a) ground infiltration into the slow storage, b) evapotranspiration and c) runoff out of the slow storage, d) percolation from the slow storage, into the baseflow storage and e) runoff out of the baseflow storage during the summer 2010. All these rates are computed on the ground areas only.





**Appendix F:  Aid to understand NSE and KGE behavior**

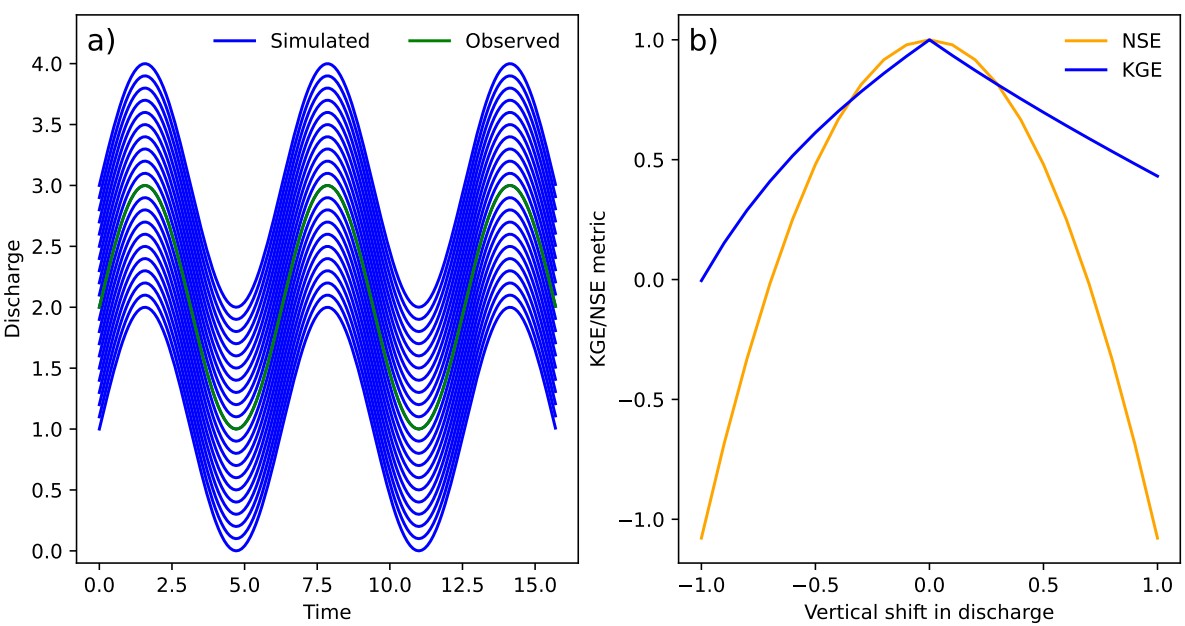

**Figure F1.** a) Vertical shift between the observed and simulated discharges, and b) the associated changes in NSE and KGE.

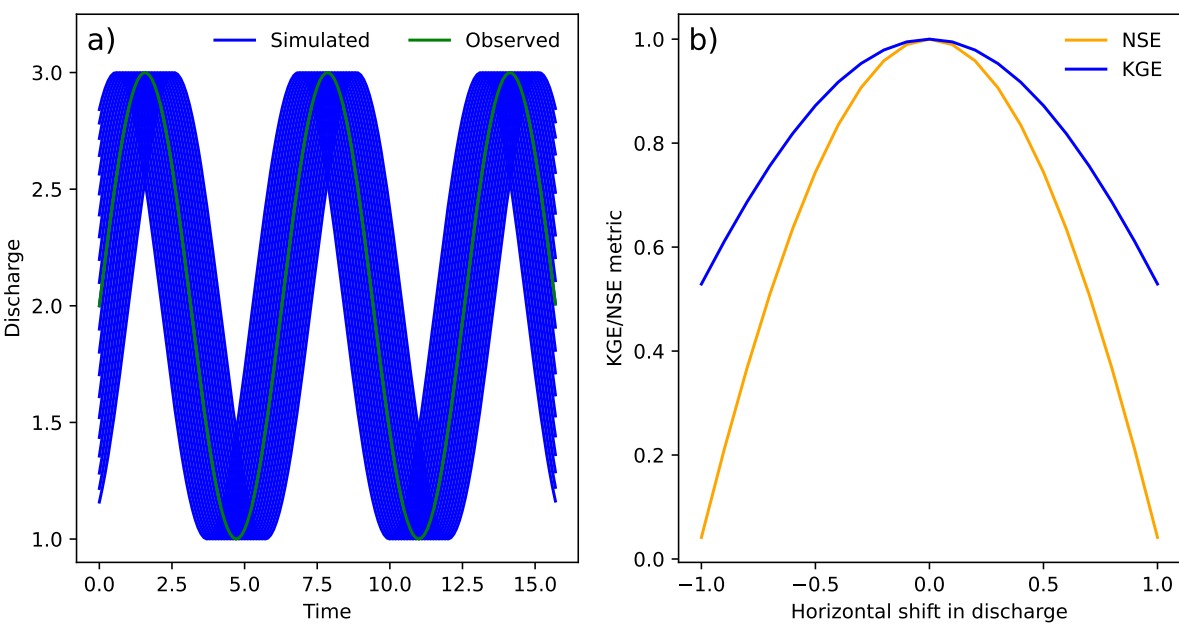

**Figure F2.** a) Horizontal shift between the observed and simulated discharges, and b) the associated changes in NSE and KGE.




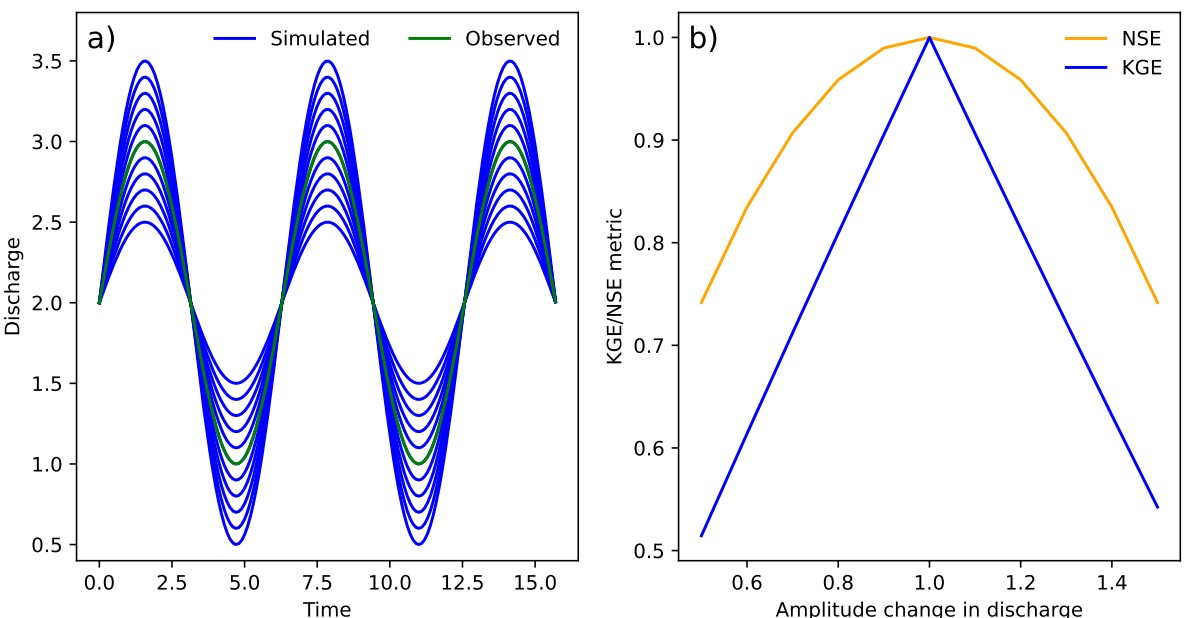

**Figure F3.** a) Amplitude change between the observed and simulated discharges, and b) the associated changes in NSE and KGE.

## 475    F1    KGE vs NSE scoring

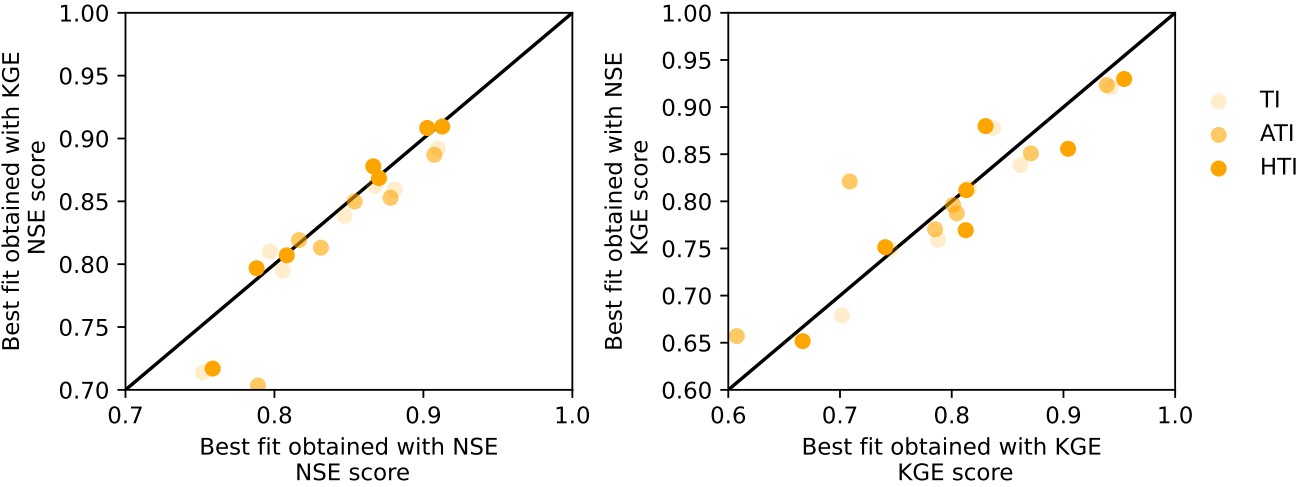

**Figure F4.** Comparison of the performance of the two NSE and KGE performance criteria in finding the calibrated parameters that are then transferred onto the different catchments. In x the NSE/KGE score when transferred with parameters obtained through NSE/KGE, rep. calibration, and in y the opposite.



*Author contributions.* Conceptualization: A.-L.A., S.L., F.C.; Methodology: S.L., P.H., B.S., A.-L.A., J.S.; Software: P.H., B.S., A.-L.A., J.S.; Validation: A.-L.A., P.H.; Formal analysis: A.-L.A., B.S.; Investigation: A.-L.A.; Resources: S.L., P.H., B.S.; Data curation: A.-L.A., P.H.; Writing—original draft preparation: A.-L.A.; Writing—review and editing: A.-L.A., P.H., B.S., J.S., F.P., L.R., M.G., S.B., S.L., F.C.; Visualization: A.-L.A.; Supervision: S.L., F.C., P.H.; Project administration: S.L., F.C., A.-L.A.; Funding acquisition: S.L., F.C.. All authors
480 have read and agreed to the published version of the manuscript.

*Competing interests.* The authors declare that they have no known competing financial interests or personal relationships that could have appeared to influence the work reported in this paper.

*Acknowledgements.* We thank Grande Dixence SA and HydroExploitation for providing the discharge data and the national weather service MeteoSwiss for providing the meteorological time series. This study was carried out in the context of the Swiss-Italian project AltroClima
485 that analyses climate change impacts on bedload transport in the Alps. The Swiss part of this project was funded by the Swiss National Science Foundation, and the Italian part by the Autonomous Province of Bozen-Bolzano, South-Tyrol. We acknowledge the use of AI to enhance the clarity of our English phrasing.



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
