# Peer review of "Scale-dependency in modeling nivo-glacial hydrological systems: the case of the Arolla basin, Switzerland"

_EGUsphere, 2024_

## Author Comment (AC1)

**Reply to RC1**

We thank the referee for their review and constructive comments. Original review comments are shown in **black,** while our replies are provided in **green**.

**Review "Scale-dependency in modeling nivo-glacial hydrological systems: the case of the Arolla basin, Switzerland" by Argentin et al.**

This study presents a modelling exercise for the small Arolla basin in Switzerland (26 km2), with the aim of finding out how model parameters vary for nested and neighboring catchments and if parameters are thus spatially transferable for a semi-distributed/semi-lumped hydrological model. The calibration of the hydrological model is done for three different melt models, two different objective functions (NSE and KGE), and with or without considering debris cover on the glaciers. The many results show that parameters are transferable, and that similar parameters can thus be used to simulate streamflow along a river network (up- and downstream). For the temperature index melt model of Hock, that includes potential solar radiation, parameters are more similar across the basins when compared to the other two melt models that use less spatial information.

Thanks for the nice summary of our work.

Overall, I found this a very interesting study with many different aspects that were looked at, a well written methodology section and a good presentation of the results and the discussion. However, I have a few points that require some further attention, which mainly relate to the framing of the study and clarity of the results.

1. Introduction – a) when reading the introduction the first time, I was a bit confused by the reasoning of why parameter transferability may be difficult in glacierized catchments and the later following explanations of the different melt models. At first, I thought that the study would focus on the different storage and routing parameters, but it turned out that the study focuses rather on the melt modelling (catchment-wide melt contribution). Although this becomes clear towards the end of the introduction, I think it should be mentioned earlier on, with an explanation of why this focus was chosen. b) What is also missing in the introduction is a discussion of studies that use glacier mass balance or snow related data to calibrate models. In such cases, melt parameters are calibrated independent of catchment size and the problems outlined here are not (less) applicable, especially if these measurements are available at large scale (remote sensing data). This is an important consideration for the framing of this study. c) And last, I think that the comparison between calibrating the model with NSE or KGE as objective function should be already mentioned as background material in the introduction. It would be good if it becomes clear how that fits too into the story of modelling glacierized catchments.

a) We agree with the reviewer and will change the introduction to clarify that we focus on the melt contribution of the different areas modeled and why. We chose this focus because storage in high alpine glacierized catchments is often shallow and plays a more minor role compared to the melt contribution.

b) In the introduction, we will add a discussion of studies using glacial mass balance and/or snow data to calibrate models. We will discuss how glacier mass balance and snow data can indeed help constrain the calibration of the hydrological model, yet it remains necessary to study the transferability of parameters between catchments to predict discharge at ungauged locations situated within the catchment.

c) We will introduce the comparison between NSE and KGE earlier in the article to clarify that our goal is primarily to assess the sensitivity to the objective function and if our conclusions still hold with another commonly used objective function.

1. Although section 4.4 is very interesting, I found the formulation and presentation a bit weak at the moment. It would help if there was already a discussion on NSE and KGE and its different sensitivities in the introduction. Furthermore, the section mixes discussion with results, although a discussion section is following. And third, I do not understand why the neighboring catchments were not included too in the analyses of Figure 12?

1) This comment ties in with the previous one, and we will discuss the sensitivities of the NSE and KGE (based on existing knowledge) in the introduction. 2) Indeed, we will move the second paragraph, which is more of a discussion paragraph, to the Discussion section. 3) We will include the neighboring catchments in the analyses of Figure 12.

2. Section 4.6 is supposed to give some insights into the physical meaningfulness of the transferred parameters, but does not discuss any of them. After reading this section I was not sure what message to take from it, besides that catchments with higher glacier cover show higher discharges later in the season. However, this is not related to the different weather patterns suggested in the lines before. Maybe this section could be tied in with the section where figure 12 is discussed?

We agree and will tie this section to the section 4.4.

3. Overparameterization – one of the conclusions of the study is that including debris cover in the calculations likely leads to overparameterization and therefore a less good fit when transferring parameters from one catchment to the other. While this may well be the case, I found the reasoning not always strong, as for example some of the melt models use more parameters (ATI) than including debris cover in the TI or HTI model. Without testing the removal of more parameters (even simpler models) and seeing when the model reaches some state of overparameterization, these conclusions are quite speculative. Maybe the authors could better argue why they choose that overparameterization might be occurring, rather than the debris melt being too spatially variable due to variable thickness?

We agree and we will modify the text to open to additional explanations on why debris cover did not improve the simulations.

4. In the last part of the discussion (5.3) I found the discussion on catchment sizes and transferability of parameters hard to follow (L412-L426). It seems to start with explaining that there is no converging of parameters for all melt models in small catchments, and that there is no improvement for the HTI model compared to the TI model, to then conclude that the HTI model parameters for catchments smaller than 7 km2 are good transferable. Please check the reasoning here and clarify. More on a general note, I was missing some discussion on the implications of the results presented in this study. For example, do the results hold for this specific set-up, or for this catchment size (26 km2 is still rather small), or can we expect similar transferability effects in larger catchments? And would the results also apply to parameters other than melt related parameters?

We will clarify this part of the discussion and will add a discussion paragraph on the implications of this study on transferability applied to larger catchments and other, non-melt related parameters.

**Detailed comments**

Thank you for the detailed comments, which we will consider during the manuscript's revision. Below we answer those comments that go beyond simple corrections:

- L8 "streamflow patterns" – what is meant here with patterns?
  We meant it in the sense "streamflow regime", but we will remove the word "patterns" to make the sentence clearer.
- L29 – L34 – maybe it would help to also add that the category of "lumped" models here, do not include a "real" spatial location, and that is the reason why they are calibrated at the outlet (as the functioning of the whole system).
  In my opinion, since most "lumped" models still take into account the characteristics of the area they are trying to model (glaciated area, debris cover area) they are still anchored in a "real" spatial location. We will however highlight that they reflect the functioning of the whole system, as requested.
- L62 what is a "parameter bias"?
  We mean it in the sense of "parameter error". We will clarify.
- L66, possibly use another symbol for degree day factor, as "a" is often also used for albedo.
  We do not think this is a problem in this paper.
- L74 what is meant with "correlation of aspects?"
  This refers to the spatial decorrelation, i.e. the distance beyond which the spatial variance does not increase anymore. We will clarify this point.
- L133-135 possibly add that the discharge data were normalized with the same maximum values? Otherwise biases between simulations and observations will not be visible.
  Yes, we will specify this "normalized by the same highest observed discharge values."
- L161 "along with baseflow from melt and rainfall" – what is meant here? The outflow from the groundwater/slow reservoir?
  We mean the baseflow originating from snowmelt or rainfall on ice-free ground and ice melt. Contrary to the baseflow originating from snowmelt, icemelt and rainfall on glacier-covered ground. We will add a reference to the Figure A1 that explains this better.
- L163 – "fed into (to – remove) two lumped parallel reservoirs" – can you briefly describe what these reservoirs represent?
  Yes, thank you, we will add that.
- L193 "night time" – at what temporal scale is Ipot calculated?
  Ipot is computed at the 15 min scale, we will add this to the manuscript.
- L202 "downscaling" – how was this done?
  We will reformulate the sentence. The downscaling is done by computing the weights representing the contribution of each data cell to each HRU based on their spatial coverage. These weights are then used to derive the mean values for each HRU, for each daily time step.
- L211 "but an inequality is added" – can this be described in a more elaborative way? How much is it, and to which parameters is it applied? Are estimates of debris thickness known to assure it has a lower melting effect instead of an increasing effect?
  We will describe this part in more detail.

- L250 "more robust yearly signatures" – this sounds like stable flow, while what is meant here is that it is more variable flow, right?
  You are right, this is wrong wording, we will reformulate.
- In Section 4.5, what is the difference between reason i and reason iii?
  We mean in i) the use of a parameter that overfits the model and might not be transferable to new data, while in iii) we refer to the impossibility to find a unique set of parameter explaining the results. These reasons are tied, and we will try to clarify this part.
- L371-372 "Longer in-stream flow paths lead hereby to a stronger dampening effect of hillslope- and glacier scale runoff variability" – Could the mechanisms behind this be explained here? Does it relate to more sub-surface storage and other streamflow components to compensate the glacier runoff variability?
  Thanks for this comment, we were referring here simply to the smoothing effect of streamflow waves traveling downstream (geomorphological dispersion; Rinaldo et al., 1991).

**Figures:**

Thank you for the detailed comments on the Figures, which we will consider. Below, we answer those comments that require an answer:

- Figure 5 – what does the distribution of parameters represent here? Are these all parameters of the 10000 simulations?
  Yes, these are the all the parameters from all the simulations.
- Figure 12 the "relative difference" units are not very clear. Maybe "relative difference" can be explained in the caption, as well as "relative performance change", is it a fraction of the original one (i.e. calibration run /BI characteristic?). And how was the over/underestimation assessed?
  We agree that the unit is not clear, we will thus rewrite the explanation as a formula as suggested. The over/underestimation was assessed visually, we will specify it.
- L446 "when focusing on attenuation of discharge trend offsets" – what is referred to here? This comes new in the conclusion and hasn't been explained before, at least not using these wordings.

  True, we will change the wording to match the places where we mentioned this.

References:

Rinaldo, A., Marani, M., and Rigon, R.: Geomorphological dispersion, Water Resources Research, 27, 513-525, 1991.

---

## Author Comment (AC2)

**Reply to RC2**

We thank the referee for their review and constructive comments. Original review comments are shown in **black** while our replies are provided in **green**.

The paper "Scale-dependency in modeling nivo-glacial hydrological systems: the case of the Arolla basin, Switzerland" investigates the transferability of parameters of a semi-lumped hydrological model within nested catchments in a high Alpine environment. It specifically explores the role of including physiographic information by implementing temperature-index models with decreasing simplicity for modelling snow and glacier melt. The authors conclude that including the effect of solar radiation in melt modelling increases the transferability of parameters, while including the effect of debris coverage of glaciers reduces parameter transferability.

The paper is well written, and presents a comprehensive modelling study and analysis that is of potential interest to the community of lumped hydrological modellers. A few concerns, however, occurred to me while reading the draft. These are outlined below and should be addressed by the authors in a revised version before the paper can be published in HESS.

The largest concern in my opinion is that the role of precipitation input is hardly discussed. The authors are making quite some efforts to explore the role of physiography for differences in production of melt water, which arguably is a major source for river discharge in their study area. At the same time, the role of physiography for differences in precipitation (snow, rain) input to the studied catchments is not analysed or discussed. The authors explain that local station data is limited and gridded data is used for precipitation, but it would be interesting to know if and how the precipitation characteristics could also explain some of the differences of the subcatchments. The focus here is on melt modelling, for which spatial differences in snow pack accumulation could be important. Similarly, also rainfall patterns could be important for shaping the discharge from different subcatchments, at least it appears that rainfall-runoff after depletion of snowpack produced some of the highest discharges in the observations (e.g., Figures 14 & E1).

I thus encourage the authors to add some analysis of the precipitation patterns, for example: Are there differences in precipitation input among the catchments? What about inter-annual variability? How do the gridded precipitation data compare to local info (at least two meteo stations are mentioned)? These issues should be discussed critically, especially if and how these relate to the parameter transferability between subcatchments.

Thank you for bringing to our attention that the current version of the paper does not pay enough attention to the role of precipitation. We currently have one graph showing the low variation between the daily precipitation in the different catchments of our study (Fig. 14a), but we will add the long-term annual precipitation trends for all catchments in the Supplement, as well as a comparison with meteorological station datasets. We will add a discussion paragraph to the article to discuss the role of precipitation.

Another doubt regards the bootstrapping approach the authors use as a benchmark. This is not critical for the evaluation of the paper, but perhaps some more explanation or even a revision would be possible. If I got it correctly, five years of observations were resampled in yearly blocks to obtain 100 benchmark series of discharge, such that each year is represented by a random choice of one of

the other four years. Goodness-of-fit of observed and resampled series are calculated and averaged. The authors find that these benchmark values drop with decreasing size of subcatchments.

While I get the idea of providing a benchmark that preserves some of its characteristics like autocorrelation, it is not entirely clear to me what the assumptions behind this specific implementation of bootstrapping as a benchmark are.

The chosen bootstrapping method was retained because it is an easy metric to compute and it gives a good idea of the fit of the model in comparison with a regime simulated based on previous years. We will develop this point in the discussion.

Would it not be problematic if the precipitation dynamics were different in different years?

It would, indeed, make the bootstrapping less meaningful for years with different dynamics. However, we argue that our catchments are all dominated by the same snowmelt / icemelt dynamics that can be seen every year, even if with some temporal variability. This is why taking 100 combinations aims to compensate for outlier years. We will develop this point in the discussion.

How similar are the resampled series to each other, given that only five yearly blocks were used?

We will show in the Supplement of the revised version how different the resampled series are (by plotting them).

Why are 100 random combinations used, and not all possible combinations?

This heuristic choice was motivated by the origin of the bootstrapping method, that has a strong random component to it. This could have been done with all possible combinations.

Does it make a difference whether the series are averaged or the goodness-of-fit criteria are averaged?

Yes! Given the definition of the criteria, there is a non-linear mapping between the daily residuals (difference of the time series) and the criteria, accordingly, the criteria of the mean is not equal the mean of the criteria.

Could averaging the discharges for the same day of year provide an alternative and possibly more robust metric?

This is indeed a possible benchmark and the one introduced by Schaefli and Gupta (2007). This benchmark is particularly interesting for long time periods where it gives a robust representation of the average seasonal signal. In our case, we compute the benchmark over the period 2010 to 2014, which is short and the resulting average of the day of the year could be strongly influenced by a single year in that period, which can be avoided with the retained bootstrapping method. We will specify this in the revised version.

What insights does the drop of benchmark metrics with catchment sizes provide for similarity of the subcatchments, for example regarding their interannual dynamics?

Thanks for this comment, which relates to the drop of the benchmark values as a function of catchment size, as discussed at the start of the discussion section. Our justification based on the geomorphology (stream order) and in-stream flow paths was not clear (see a comment on the same point by reviewer 1), we will further elaborate on this. We will discuss better why the interannual

streamflow variability is higher at smaller scales and also discuss the hydrological similarity between the catchments.

Furthermore, we will clarify that the bootstrapped series have each a length of 5 years, which was not clear from the current manuscript.

*Further comments*

Thank you for the detailed comments, which we will consider during the manuscript's revision. Below we answer those comments that go beyond simple corrections:

130-131: Does analysing the time lags between the hydrographs support your assumption?

Yes, for example, we found a time lag of about 15 minutes between the hydrograph of BI and the hydrograph of its biggest contributor, HGDA, (we took the 1$^{rst}$ of August 1985), which is negligible at the daily scale.

Fig. 8: It appears as if the maximum discharge of 1.0 was never captured by the model. What are the reasons for this discrepancy?

The maximum discharge of 1.0 in Figure 8 is reached by the observed dataset of BI, in June. All the discharge datasets in this plot are divided by this value, which explains why the other lines do not reach 1.0. This peak of discharge is related to a Foehn event (lines 259-261) that melted the snow but could not be captured efficiently by the model, probably due to a partial record of the temperature in the gridded input values or to the action of the wind, which is not accounted for in our model. We will explain the normalization and discuss this Foehn event further in the manuscript in link with the figure.

315-316: What about the spatial variations in precipitation? Can these also be highly variable?

Thanks, we will discuss this in the revised version.

372-375: I agree with that statement, but this directly invalidates your benchmarking approach, doesn't it? "In contrast, even simple meteorology-based hydrological models deliver much better results" – so what would be the best option in the end?

This comment refers to the following sentences in the discussion: "Longer in-stream flow paths lead hereby to a stronger dampening effect of hillslope- and glacier-scale runoff variability. Given the inherent year-to-year variability in meteorological patterns, and the close link between meteorology and discharge, it ensues that in small catchments, the discharge patterns from previous years are poor predictors of the current discharge. In contrast, even simple meteorology-based hydrological models deliver much better results"

We see your point – an ideal benchmark should not depend on scale. But we do not see at this stage how to construct such a benchmark. We will make this clear in the revised version.

386: "As discussed previously" – maybe I missed it, but where was explained why simulated hydrographs should outperform NSE and match KGE?

This was not clear, we refer to the above lines "As a result, the NSE is much more sensitive to changes in bias, changes in variability or shifted yearly patterns than the KGE (see Supplementary

Material, F; Knoben et al., 2019). Thus, the benchmark KGE is a much harder criteria to meet for simulated discharges than the benchmark NSE".

We will add the following sentence to this paragraph to make the transition smoother: "We thus expect the simulated hydrographs to outperform the benchmark NSE and match the benchmark KGE."

399-404: If there were clear relationships of discharge and physiography - would taking them into account explicitly in your model solve a part of the transferability problem? This could hint at structural deficits of the model.

Yes we agree and will elaborate on this in the revised version.

References:

Horton, P., Schaefli, B., Hingray, B., Mezghani, A., and Musy, A.: Assessment of climate change impacts on Alpine discharge regimes with climate model uncertainty, Hydrological Processes, 20, 2091-2109, 10.1002/hyp.6197, 2006.

Schaefli, B., and Gupta, H.: Do Nash values have value?, Hydrological Processes, 21, 2075-2080, 10.1002/hyp.6825, 2007.

---

## Referee Report (RR1)

**Revision**

I thank the authors for their replies and revisions, which helped to clarify some of the doubts raised in the first review.

My first concern was about the heterogeneity of the precipitation input, which is important in the context of parameter transferability, because the goodness-of-fit is not only determined by the appropriateness of the calibrated parameters, but also of the input data. I found the additional info you provided on the precipitation helpful. The data of the four observation stations show a very clear trend of precipitation amounts in east-west direction, and the input data you are using seem to reflect these differences, as you rightfully point out in the discussion.

Lines 391-393: I would, however, like to question why you think that calibration on a five-years period reduces the impact of rainfall patterns on transferability to neighbouring catchments? I do not see how calibrating on the period 2010-2014 would solve this problem, especially if the difference in rainfall patterns is consistent over time as indicated by the station data shown in the Appendix. Maybe consider deleting this statement.

Would it not be better to just state that an impact is likely, and leave it like that? You are focussing on snowmelt, so it should be fine just to mention that effects from rainfall variability that are not captured in the input data are possible.

The next point was the benchmarking approach. In light of the additional explanation provided in the reply to the review and the related revision, several concerns remain, which I would like to point out in the following.

The benchmark is used to assess the quality of the parameter transfer, thus it should be unbiased to provide a fair evaluation. I think that this is not the case with the approach taken by the authors.

Excluding the data for a particular year distorts the randomness of the benchmarking approach. This design of the benchmark prevents a good fit, in the sense that a good fit cannot even be randomly achieved. This means that preventing the random draw of the original year at its original position systematically decreases the NSE and KGE values. The authors claim that "The benchmark NSE and KGE correspond to the prediction potential of the discharge dataset itself." This is not true, because due to the exclusion it is not a randomly drawn sample.

I also do not see how the approach could possibly reduce the effect of outlier years. Either there are "outliers" present in the sample of size 5, or not. If there are outliers, the NSE and KGE will be low, because an outlier year would be represented by an average year, or the other way round. I thus agree with your statement that the different years should not be too different for the approach to be meaningful (line 409). On the other hand, if the discharge series would be very similar each year, the approach also does not make much sense, it would just deliver perfect fits. As you are showing schematically in the Annex, small deviations (small in terms of the temporal resolution) in the discharge already can let NSE or KGE drop, even if the general annual pattern is similar. The question is, how much difference in dynamics is allowed and how much is required for this metric to be useful?

What also puzzles me is the limitation to five years. Discharge data is available since 1971 (line 130), the calibration period was 2009-2014, but the benchmark was only calculated for the "evaluation period" 2010-2014, which is shorter because the first year was discarded. Why do you not use the

entire dataset for bootstrapping, or averaging? Limiting to five years seems to be an unnecessary restriction. Expanding the data set would also mitigate the problem of possible outlier years.

I am not sure if a benchmark is needed at all, or if the results of the study would also be valid without this comparison. The authors state that NSE and KGE have no absolute meaning (lines 263-264). But I would claim that NSE and KGE have defined properties with regard to the statistics of the sample. Many papers have explored these metrics in the context of hydrological modelling, so they may be useful on their own because the target audience can interpret them, and their changes in the transfer experiment. The properties of the chosen benchmarking approach, however, remain unknown.

In the revision, the authors added a new section to the beginning of the discussion named "A new benchmark". This claim appears a bit bold to me, and seems to be out of scope . In my opinion, introducing a new benchmark would require a convincing approach to start with, proper testing, and critical discussion in light of the existing literature, all of which I cannot see here. From the manuscript it is not clear if the authors consulted any of the existing literature on methods for bootstrapping of dependent data and their requirements, for example in terms of sample size or the effect of excluding data while drawing.

All of the above raises a major concern if the metric is a fair benchmark, or if it is designed to give results that make comparisons against it look good. I suggest considering an unbiased benchmark, either by one of the suggestions above or something that is described in the literature, or getting rid of the benchmark entirely.

Minor:

- The Figure G is meant to show that the relative impact of precipitation is small vs. snowmelt. The time period is not indicated, so the comparison with Figure F is difficult and it is hard to tell if this is a larger event, or if a minimal rain event was chosen.
- Figure F - I suggest that you add the relative position of the observation stations to the figure caption of F to give a better idea (like: "xyz km to the NE of Arolla station").
- Line 419: "predictability of discharge from past discharge signals" – your benchmark consists not only of past years, but "other" years.
- Lines 433-437: *"Given the inherent year-to-year variability in meteorological patterns, and the close link between meteorology and discharge, it ensues that in small catchments, the discharge patterns from previous years are poor predictors of the current discharge. In contrast, even simple meteorology-based hydrological models deliver much better results. An ideal benchmark should not depend on scale; however, we do not see at this stage how to construct such a benchmark."* – Again, I would like to point out that this seems contradictory. If simple meteorology-based hydrological models give good results, why not use them as a benchmark? Maybe I am missing a point here, but perhaps the last two sentences are not really needed, and could be skipped?